

# The Impact of Seismic Interpretation Methods on the Analysis of Faults: A Case Study from the Snøhvit Field, Barents Sea

*Jennifer Cunningham\*[1,3], Nestor Cardozo[1], Chris Townsend[1], Richard Callow[2]*

[1]*Department of Energy Resources, University of Stavanger, 4036 Stavanger, Norway*

5          [2]*Equinor ASA, Forusbeen 50, 4035 Sandnes, Norway*

[3]*Equinor ASA, Sandslivegen 90, 5254 Sandsli, Norway*

*\*+47 461 84 478, jenecunningham@gmail.com*

**Keywords:** seismic interpretation, fault length, throw, juxtaposition, geological modelling, petroleum volumes, Snøhvit field

**Abstract**

Five seismic interpretation experiments were conducted on an area of interest containing a fault relay in the Snøhvit field, Barents Sea, Norway, to understand how interpretation method impacts the analysis of fault and horizon morphologies, fault lengths, and vertical displacement (throw). The resulting horizon and fault interpretations from the least and most successful

interpretation methods were further analysed to understand the impact of interpretation method on geological modelling and hydrocarbon volume calculation. Generally, the least dense manual interpretation method of horizons (32 inlines (ILs) x 32 crosslines (XLs), 400m) and faults (32 ILs, 400m) resulted in inaccurate fault and horizon interpretations and underdeveloped relay morphologies and throw that can be considered inadequate for any detailed geological analysis.

The densest fault interpretations (4 ILs, 50m) and auto-tracked horizons (1 IL x 1 XL, 12.5 m) provided the most detailed interpretations, most developed relay and fault morphologies and geologically realistic throw distributions. Analysis of the geological modelling proved that sparse interpretation grids generate significant issues in the model itself which make it geologically inaccurate and lead to misunderstanding of the structural evolution of the relay. Despite

significant differences between the two models the calculated in-place petroleum reserves are broadly similar in the least and most dense experiments. However, when considered at field-





scale the magnitude of the differences in volumes that are generated solely by the contrasting interpretation methodologies clearly demonstrates the importance of applying accurate interpretation strategies.

## 1. Introduction

An accurate understanding of faults in the subsurface is critical for many elements of the hydrocarbon exploration and production industry. For example faults control sediment and reservoir depositional systems, act either as conduits or baffles to fluid flow, are the defining elements of structural traps and impact the design of exploration and production wells (e.g.

Athmer et al., 2010; Athmer and Luthi, 2011; Botter et al., 2017; Fachri et al., 2013a; Knipe, 1997; Manzocchi et al., 2008a, 2010). Subsurface faults are commonly interpreted on either reflection seismic data or attributes of that data by creating fault sticks on vertical cross sections (e.g. inlines ILs or crosslines XLs) which are then used to generate fault surfaces. Fault displacement is analysed by studying the interaction between the displaced horizon reflectors and

the fault surface. Although this is a commonly used interpretation method, the impact of changing interpretation density (i.e. inline or crossline spacing), interpretation on vertical vs horizontal sections, and the effects of manual vs auto tracking techniques have not been systematically investigated.

The interpretation of faults in seismic data has been the focus of many studies. Badley et al.
(1990) were the first to publish a systematic approach to the seismic interpretation of faults using fault displacement analysis and horizon correlations across multiple intersections. Freeman et al (1990) explained how fault displacement analysis can be used in the quality control process of fault analysis. The interpretation of fault surfaces has been quality controlled by projecting longitudinal and shear strain (vertical and horizontal components of dip separation gradient) onto

fault planes and assigning realistic strain limits in order to identify inaccurate interpretations (Freeman et al., 2010). Uncertainty in fault interpretation has also been readily analysed and previous works have focused on how significant uncertainties and interpretation biases exist in 2D and 3D seismic interpretation (Bond, 2015; Bond et al., 2011, 2007; Schaaf and Bond, 2019) and the impact of the image quality of seismic data on uncertainty in seismic interpretation

(Alcalde et al., 2017). Uncertainty pertaining to fault properties and the effect fault properties



have on fluid flow simulations have also been analysed (Manzocchi et al., 2008b; Miocic et al., 2019).

Many techniques have extended basic fault interpretation techniques in order to better understand the link between faults in seismic and their properties in the subsurface. Dee et al. (2005) studied the application of structural geological analysis to a number of common industry based techniques and workflows (e.g. fault seal, fluid accumulation, migration, fault property modelling). Seismic attributes have been analysed to study fault architecture and investigate fault sealing potential (Dutzer et al., 2010). Long and Imber (2010, 2012) used interpreted seismic surfaces to measure regional dip changes in order to map fault deformation in both a normal fault array and a relay ramp. Studies such as these, combined with the increasing availability of high-resolution 3D seismic data have driven seismic structural analysis towards more detailed and quantitative studies. Iacopini and Butler (2011) and Iacopini et al. (2012) generated a workflow combining seismic attribute visualization with opacity filtering and frequency decomposition to characterize deep marine thrust faults. In a case study from the Snøhvit field, a linkage between unsupervised seismic fault facies and fault related deformation was established and seismic amplitude was analysed to understand how folding near faults might influence near fault amplitudes (Cunningham et al., 2019).

Synthetic seismic modelling has shed important light on the impact of seismic frequency on fault imaging, the seismic amplitudes contained in and around faults and their linkage to fault related deformation and fault illumination (Botter et al., 2014, 2016b, 2016a). A comparison of faults in the Snøhvit field with synthetic seismic showed the importance of incidence angle, azimuthal separation and frequency on fault imaging (Cunningham et al., submitted).

Fluid flow across faults, through deformed bedding and the sealing properties of faults have long been important topics in the petroleum industry (e.g. Bretan et al., 2011; Caine et al., 1996; Cerveny et al., 2004; Davatzes and Aydin, 2005; Edmundson et al., 2019; Fachri et al., 2013b, 2013a, 2016; Fisher and Knipe, 1998; Knipe, 1997, 1992; Yielding et al., 1997). In addition, reservoir modelling techniques have been used to simulate fluid flow across faults (Fachri et al.,





2013a), and synthetic seismic modelling has been used to understand the impact of faulting and fluid flow on seismic images (Botter et al., 2017).

Fault interpretation in seismic data has formed the basis of many studies over the decades but no single study has looked specifically into seismic interpretation methodologies. It would seem logical to assume that increased interpretation density will result in a higher resolution output (i.e. fault and horizon interpretation), but at the expense of the increased time required to perform the interpretation. It has yet to be fully evaluated, whether these more detailed interpretations

justify this increased time and effort, or whether the end results are comparable to much more efficient interpretation strategies. Similarly, auto-tracking algorithms would appear to offer a shortcut to high-resolution horizon and fault interpretations, but how do these algorithms compare to the results of detailed manual interpretations? We address the impact of interpretation strategy on the quality of the final products and whether it is possible to identify an

optimum balance between interpretation density, time required to do the interpretation, and the accuracy of the end-result.

Our study will test the effect of interpretation method (faults and displaced horizons) on aspects of fault analysis with the aim to provide geoscientists with a better knowledge of seismic interpretation/analysis of faults and an explanation of the implications of improper interpretation

and best practice interpretation methods. We have designed five fault and horizon interpretation experiments which were conducted on a seismic volume case study from the Snøhvit field, Barents Sea. The resulting surfaces from each experiment (faults and horizons) were run through a fault analysis workflow. Key aspects of the workflow include the analysis of: fault length and morphology, displacement of both faults (throw; Badley et al., 1990; Freeman et al., 1990),

juxtaposed lithology (Allan, 1989; Fisher and Knipe, 1998; Knipe, 1992, 1997), dip separation gradients (Freeman et al., 2010), and finally geological modelling (i.e. Turner, 2006) and the subsequent petroleum volume calculations.

## 2. Geologic Setting

The Snøhvit gas and condensate field is located in the centre of the Hammerfest Basin on the

southwest margin of the Barents Sea (Fig. 1a, b: Linjordet and Olsen, 1992). The ENE-WSW trending Hammerfest basin is ~150 km long by 70 km wide and is bound in the north, southeast and west by the Loppa High, Finnmark Platform and Tromsø Basin respectively. Rifting in the



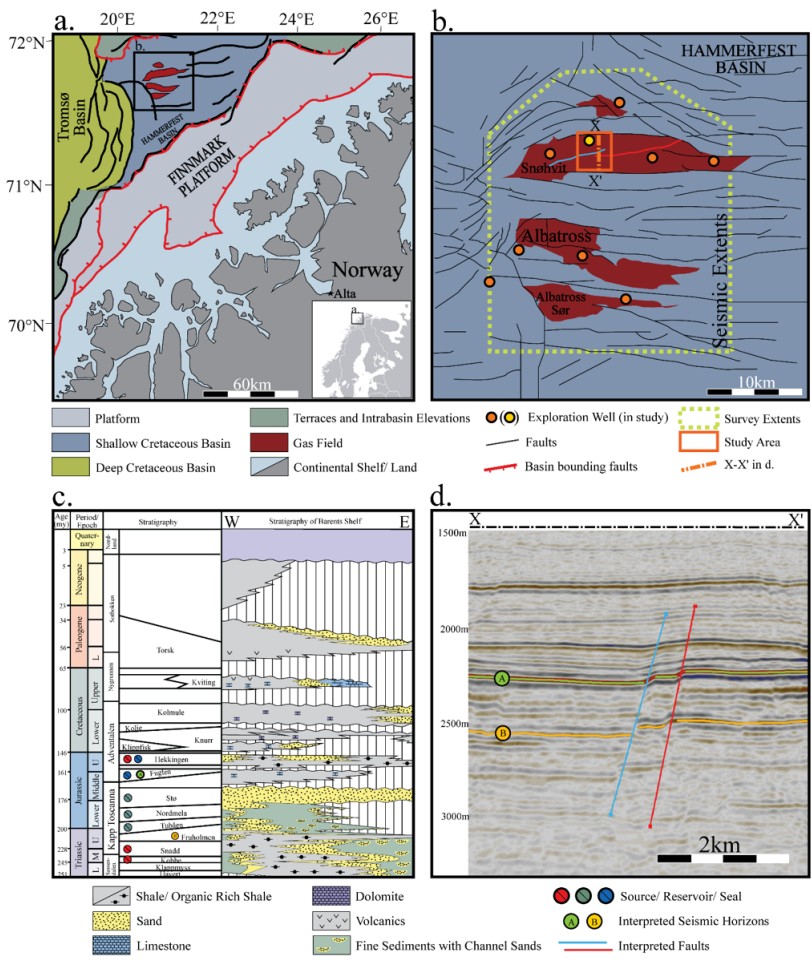

Figure 1: (a) Geologic setting of the Hammerfest Basin. The area in b is marked by a black box. Modified from NPD Fact maps. (b) Snøhvit Field area. The dashed yellow line shows the extent of seismic data and the orange rectangle highlights the study area. Map modified from Ostanin et al. (2012). The blue background refers to the Jurassic Hammerfest Basin while the red shapes identify the areal extent of Lower-Middle Jurassic gas fields. The western and eastern fault in the study area are coloured blue and red respectively. (c) Generalized lithostratigraphic column of the Barents Sea highlighting the horizons of interest. Modified from Ostanin et al. (2012). (d) North-south seismic IL (3342) through the middle of the Snøhvit Field (X-X' in b) with interpreted horizons and faults. Interpreted horizons are A: Top Fuglen, and B: Top Fruholmen (c, d).



basin initiated in the Late Carboniferous-Early Permian and drove the formation of the NE-SE

trending basin bounding faults (Gudlaugsson et al., 1998). A second phase of rifting in the Late

Jurassic-Early Cretaceous reactivated the basin bounding faults and caused the basin to undergo

large amounts of subsidence on both the northern and southern margins (Doré, 1995; Linjordet

and Olsen, 1992; Ostanin et al., 2012; Sund et al., 1984). Due to differential subsidence during

this period, the Hammerfest Basin widened and deepened westward, allowing for the

accumulation of thicker sediment packages in the west (Linjordet and Olsen, 1992). A dome at

the basin's central axis and a subsequent east-west trending fault system formed during basin

extension in the Early Jurassic- Barremian (Sund et al., 1984). These east-west trending faults

define the structure of the Snøhvit field and divide the field into northern and southern petroleum

provinces (Sund et al., 1984). The main petroleum system components of the Snøhvit field are

located within the Upper Triassic-Jurassic strata (Fig. 1c; Linjordet and Olsen, 1992). The focus

of this study is in two of the east-west trending faults across the Snøhvit field (Fig. 1b, blue and

red lines). These two faults dip to the north, offset the Jurassic strata, and form a relay ramp

structure (Fig. 1d). The area was chosen because relays are structurally complex and require

special attention in their interpretation. Relays are also important in petroleum systems as they

can create sediment distribution pathways, enable or disable fault seal (as all faults can), act as

fluid flow pathways and finally can be a part of trap definitions (Athmer et al., 2010; Athmer and

Luthi, 2011; Botter et al., 2017; Fachri et al., 2013a; Fossen and Rotevatn, 2016; Gupta et al.,

1999; Knipe, 1997; Peacock and Sanderson, 1994; Rotevatn et al., 2007).

### 3.  Methodology:

Five interpretation experiments (Exps 1-5) were designed to test the impact of different seismic

interpretation methods on the analysis of faults (Fig. 2). Each of these experiments (Fig. 2a) was

completed on a chosen 5 x 5 km area covering the relay ramp (orange rectangle in Fig. 1b) and a

fault analysis workflow was applied to the interpreted seismic horizon and fault surfaces from

each experiment (Fig. 2b). The fault analysis workflow (Fig. 2b) integrates a comparison of

seismic interpretation results and analyses of fault length, throw, dip separation gradients

(longitudinal and shear strain), juxtaposed lithology, geological modelling and a calculation of

hydrocarbon volumes. While the individual components of the fault analysis workflow have





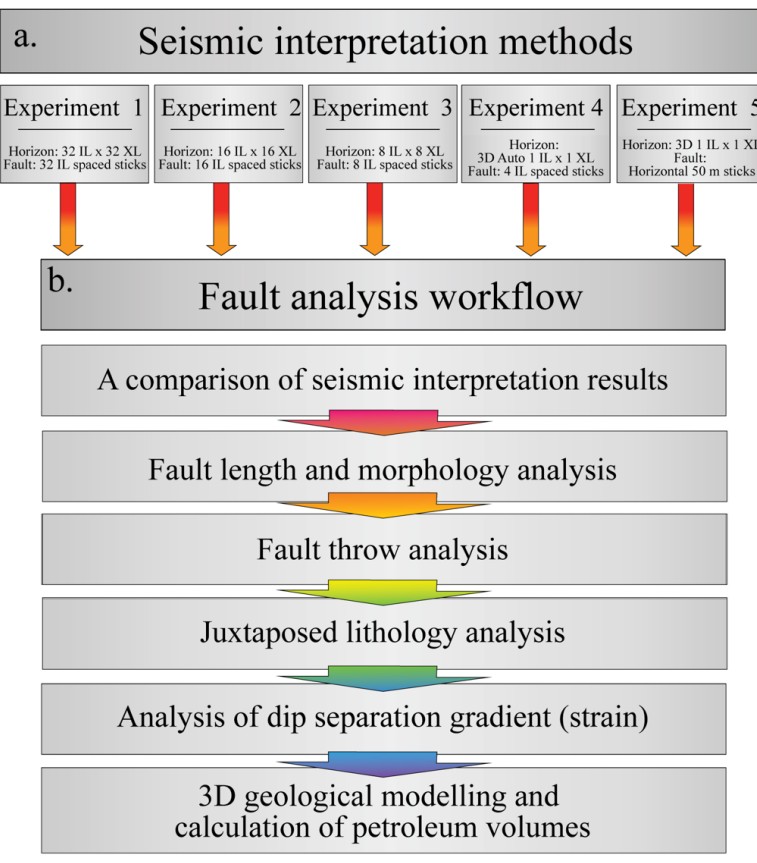

Figure *2*: The workflow used in this study. The fault analysis workflow (b) is completed on each of the seismic interpretation experiments (a).



been applied previously (e.g. Elliott et al., 2012; Fachri et al., 2013a; Long and Imber, 2010,
2012; Rippon, 1985; Townsend et al., 1998; Wilson et al., 2009, 2013), no earlier studies have
considered the impact of the seismic interpretation strategy on the outcomes of the fault analysis
workflow in its entirety.

The computer programs Petrel™ and T7™ (formerly Trap tester™) were used in the seismic
interpretation and fault analysis workflows respectively. The seismic dataset used in this study is
survey ST15M04, a merge of five 3D seismic streamer surveys that was provided by Equinor
ASA and their partners (Petoro AS, Total E&P Norge AS, Neptune Energy AS, and Wintershall
DEA AS) in the Snøhvit field, Norwegian Barents Sea. The ST15M04 volume was zero phase
pre-stack depth migrated (PSDM, Kirchhoff), and both partial and full offset stacks were
available. It is assumed that the velocity model used in the PSDM was correct and that the
vertical scale of the processed volume (in depth) represents depth in meters. The inlines (ILs)
and crosslines (XL) are spaced at 12.5 m and an increase in acoustic impedance is represented by
a red peak (blue-red-blue). The interpretation was performed in depth to give the most
representative view of the geological and structural relationships and to avoid re-stretching the
data back into time. All five interpretation experiments were conducted on the near stack data (5-
20°) as this dataset has been proven to give the most consistent fault imaging and best reflector
continuity (i.e. Shuey 1985). As the data is a merge of multiple datasets and vintages, the
acquisition orientations geometries could not be considered although they are known to impact
fault imaging (Cunningham et al., 2020).

### 3.1 Seismic interpretation

The two east-west trending, north dipping faults that form the relay ramp were interpreted (Fig.
1b, d). These two faults are termed the western and eastern faults (Fig. 1b and d, blue and red
fault respectively). Two faulted seismic reflectors (top Fuglen and Fruholmen formations; Fig.
1c-d) were also interpreted. These reflectors were chosen because the top Fuglen is a very strong,
easily interpreted reflector while the top Fruholmen is poorly imaged and is more challenging to
interpret. Both the top Fuglen and top Fruholmen are peaks (increases in acoustic impedance).
The Stø Formation, which falls between the tops Fuglen and Fruholmen, is a prolific petroleum
reservoir. Five different seismic interpretation methods (Exps 1-5) were used with the aim to
systematically study how seismic interpretation techniques (Fig. 2a) influence the fault analysis



workflow (Fig. 2b). The first three experiments are manual horizon interpretation techniques

with different IL and XL spacing (from every 8 to 32 lines), while the fourth and fifth

experiments are a combination of automated (3D auto-tracked horizons) and manual fault

interpretations.

### 3.1.1 Exp 1: 32 x 32

The top Fuglen and top Fruholmen reflectors were interpreted on a 32 x 32 IL (north-south) and

XL (east-west) grid using 2D auto-tracking (Fig. 3a). Fault sticks were interpreted perpendicular

to the average strike of the faults on every 32$^{nd}$ IL and are interpreted as largely planar features

(Fig. 3a, faults). The IL/XL spacing in this experiment is equal to 400 m (32 x 12.5 m).

The interpretation of the two horizons and the two faults took the least amount of time when

compared to all other experiments because of the large IL/XL spacing (Fig. 3a, relative time).

Overall, this experiment was the quickest but sparsest interpretation method. Since the

interpretation was manually conducted on an IL and XL basis there was no QC needed for the

top Fuglen due to the high quality of this reflector. In particularly dim areas, 2D auto-tracking of

the top Fruholmen required more manual input and some QC.

### 3.1.2 Exp 2: 16 x 16

The two horizons were interpreted on a 16 x 16 IL and XL grid using 2D auto-tracking of the

peaks for each reflector (Fig. 3b). Fault sticks were interpreted on every 16$^{th}$ IL and are largely

planar (Fig. 3b, faults). The IL/XL spacing in this experiment is equal to an interpretation

spacing of 200 m (16 x 12.5 m).

The interpretations of both the horizons and faults in this experiment took twice the amount of

time of Exp 1, since the IL/XL spacing is half. This experiment was ranked the second most time

consuming and the second sparsest overall (Fig. 3b, relative time). Since the interpretation in this

experiment is manual, a similar level of QC was needed. There is high to lower confidence in the

interpretation quality of the top Fuglen and top Fruholmen reflectors, as described in Exp 1.




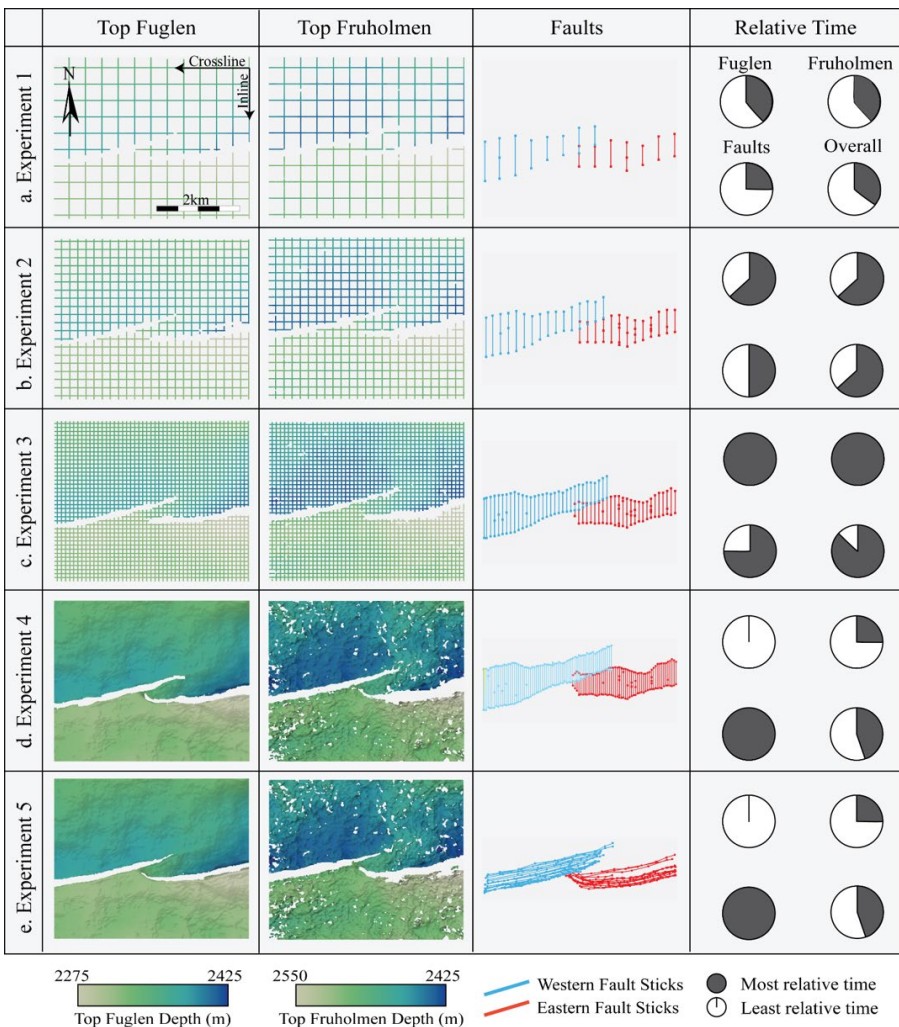

Figure 3: The seismic interpretation methods for experiments 1-5. (a) Exp 1: 32 x 32 IL and XL interpretation spacing, fault sticks are interpreted on 32 spaced ILs. (b) Exp 2: 16 x 16 IL and XL interpretation spacing, fault sticks are interpreted on 16 spaced ILs. (c) Exp 3: 8 x 8 IL and XL interpretation spacing, fault sticks are interpreted on 8 spaced ILs. (d) Exp 4: 3D auto-tracking is used to interpret horizons on all ILs and XLs, fault sticks are interpreted on every 4th IL. (e) Exp 5: 3D auto-tracking is used to interpret horizons on all ILs and XLs, and faults are interpreted on depth slices of the tensor attribute at a spacing of 50 m. Experiment related time estimations for the top Fuglen, top Fruholmen, the two faults and the average time taken for each experiment are displayed in the last column for each experiment.



### 3.1.3 Exp 3: 8 x 8

The two horizons were interpreted on an 8 x 8 IL and XL grid (Fig. 3c). Fault sticks were interpreted on every 8[th] IL (Fig 3c, faults). The IL/XL spacing in this experiment is equal to an interpretation spacing of 100 m (8 x 12.5 m).

The horizons and faults in this experiment took approximately three times longer to interpret than Exp 1. This experiment was the densest of the manual interpretation methods (experiments 1-3) and was therefore the most time consuming (Fig. 3c, relative time). The quality control and interpretation confidence of the two reflectors is as described for Exps 1 and 2.

### 3.1.4 Exp 4: 3D tracked method with dip-parallel fault sticks

Horizons were tracked using the 3D auto-tracking algorithm in Petrel™, which resulted in complete interpretation coverage for the top Fuglen compared to almost complete coverage for the top Fruholmen (Fig 3d, tops Fuglen and Fruholmen). Initially we planned to apply a 3D automated fault interpretation method (Adaptive Fault Interpretation; Cader, 2018) for this experiment but the algorithms currently available do not provide geologically realistic fault

sticks that could be used in our workflow. As a result, fault sticks were interpreted on every 4[th] IL to capture the densest and most geologically realistic morphologies possible (Fig 3d, faults). The IL/XL spacings of horizon and fault interpretations in this experiment are 12.5 and 50 m respectively.

The 3D auto-tracked interpretation of the top Fuglen was the fastest method as the reflector is

well imaged. The top Fruholmen was a little slower to run through the auto-track due to its poor seismic imaging (Fig. 3d). As a result, the top Fruholmen required more manual guidance for the auto-track to be successful but was still faster than all three manual interpretation methods (Exps 1-3). The fault interpretation for this experiment was the most time consuming as the spacing of fault sticks was the densest (4 ILs, 50 m). Overall, Exp 4 was tied for the second fastest to

interpret (Fig 3d, relative time) but also contains the highest density of interpretation lines for both the horizons and faults. The QC of the top Fuglen was completely unnecessary in this small study area as the reflector was strong and easily auto tracked. The QC of the top Fruholmen was more important since the reflector imaging is quite poor in some areas. The interpretation



confidence for this case is high to moderately high for the top Fuglen and top Fruholmen

respectively.

### 3.1.5 Exp 5: 3D Auto-tracked horizons with horizontal (strike parallel) fault sticks

This experiment used the same 3D auto-tracked horizons as discussed in Exp 4 (Fig. 3e, tops).
However, faults were manually interpreted horizontally on depth slices spaced every 50 m
vertically using the tensor attribute to guide the interpretation (Fig. 3b, bottom). The tensor

attribute is generated using a symmetric and structurally-oriented tensor which detects the
localized reflector orientation and is sensitive to changes in both the amplitude and continuity of
the seismic reflectors in question (Bakker, 2002). This attribute was chosen as it is a well-known
fault enhancing attribute and is widely used in fault interpretation (e.g. Botter et al., 2016b;
Cunningham et al., 2019). The resulting fault sticks have a high degree of horizontal curvature as

each stick traces a fault's entire lateral extent. Although the results have the same fault
morphology to Exp 4, the horizontal fault sticks look quite different to the planar dip-parallel
fault sticks (Fig. 3e, Faults).

The fault interpretation for this experiment was time consuming as it required the generation of a
tensor attribute prior to interpretation. Once the attribute was produced, the time to generate the

fault interpretation was in the middle range of the time used for the other experiments. The
interpretation confidence of the two reflectors are as described in Exp 4.

### 3.1.6  A comparison of horizon and fault surface grids

The horizons interpretations and fault sticks were gridded into horizons and fault surfaces using
the seismic 12.5 m grid spacing. The horizon surfaces were generated to stay true within 5 m of

the interpretations for each of the five experiments and no post-processing smoothing techniques
were applied to the horizon gridding. Fault sticks in all five experiments were made into surfaces
using a 50 m triangulated surface algorithm. This method was chosen as it generated a surface
that was closest to the original fault stick interpretations. The fault and horizon surfaces were
used as the input for the fault analysis workflow.

To understand the relative differences between the horizons from each experiment, thickness
maps were generated between the most densely interpreted 3D auto-tracked horizons (Exps 4
and 5) and the horizons generated from each of the manual based experiments (Exps 1-3).



Anywhere where there is a good correlation between the auto-tracked and manual surfaces, there is very little or no thickness change, while in the case of a poor correlation, a greater range in
thickness may result.

### 3.2 Fault length and morphology

Fault length (Fig. 4a) is defined as the maximum horizontal distance of a fault in three dimensions (Peacock et al., 2016; Walsh and Watterson, 1988). An analysis of fault length was conducted on the western and eastern faults (Fig. 1b, d) using the gridded fault surfaces. These
data were extracted from the edge of the study area to the fault tipline for both faults. The data were graphically compared to understand the impact of interpretation method on fault length.

To analyse fault morphology, the horizon surfaces described in Sect. 3.1.6 were used. In creating the surfaces, all horizon interpretations that fall within the fault polygons were removed, leaving behind a gap in the surface where the faults' extent and morphology through that horizon are
clear. These fault polygons were generated using patch and trim distances, this is explained in detail in Sect. *3.3 Fault Throw*. The analysis of morphology considers these voids in the horizon surfaces. The graphical representations of fault throw (upcoming in Sect. 3.3) can also be used to understand fault length.

### 3.3 Fault throw

Fault throw is defined as the vertical component of dip separation on a fault (Fig. 4a). Fault throw along the length of an isolated fault typically follows a trend where the highest throw occurs in the centre of the fault and progressively decreases towards the tip lines (Barnett et al., 1987; Walsh and Watterson, 1990; Fig 4a, inset). In this study, a separate fault throw analysis was created on each of the five experiments. To calculate throw, hangingwall and footwall
cutoff-lines were produced for the top Kolje, top Fuglen and top Fruholmen in each experiment using patch and trim distances on both faults of 150 and 75 m respectively (Fig 4b). These deal with the poor seismic image close to the fault: horizon data within the trim distance are rejected, while those within the patch distance are used to extrapolate the horizon on to the fault (e.g. Elliott et al., 2012; Wilson et al., 2009, 2013). The top Kolje (Fig. 1c) was used only to help in
any lithological projections in the sections to follow. This younger horizon is only partially folded at the western margin of the western fault, so it is not discussed further with respect to


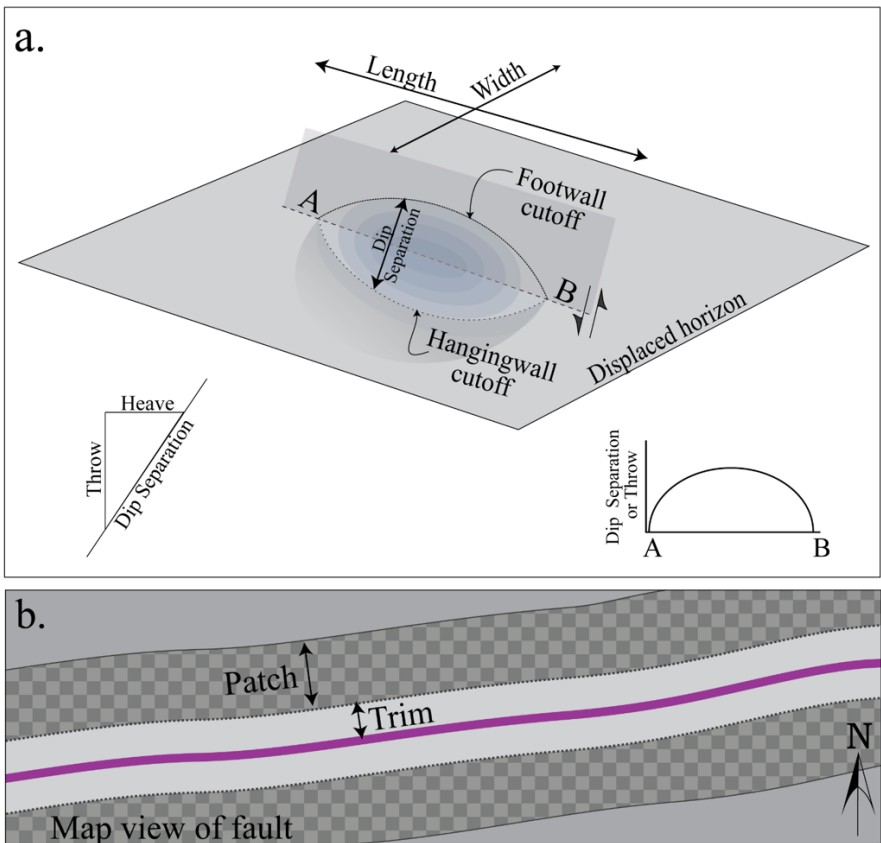

Figure 4: Fault schematic and fault throw methods. (a) 3D diagram of an isolated normal fault showing the field of displacement, hangingwall and footwall cutoff-lines, fault length and width, dip separation, throw and heave. (b). Map view of a fault with trim and patch distances used in the determination of hangingwall and footwall cutoff-lines. The patch and trim distances used in this analysis were 150 and 75 m respectively. Concepts in this figure are based on findings from Barnett et al., 1987; Elliott et al., 2012; Rippon, 1985; Walsh and Watterson, 1987, 1988; Watterson, 1986; Wilson et al., 2009.



deformation. The cutoff-lines and their dip separation were then used to calculate the throw
across the fault surface (Fig. 4a, bottom left inset). The results were displayed directly on the
fault plane and were also graphically represented to understand how fault throw changes across
each of the experiments.

### 3.4   Dip separation gradient and strain

The dip separation gradient, and the longitudinal and shear strains are useful tools for QC
seismic interpretations (Freeman et al., 2010). The dip separation gradient was calculated using
the top Kolje, top Fuglen and top Fruholmen cutoff-lines. The longitudinal strain (also known as
the vertical gradient) is the dip separation gradient in the direction of fault dip while shear strain
(horizontal gradient) is the dip separation gradient along the strike of the fault (Freeman et al.,
2010; Walsh and Watterson, 1989). In this study, we use the principles introduced in Freeman et
al. (2010) to analyse these measurements. This can help us to understand how the different
seismic interpretations strategies produce results that differ from what is considered geologically
realistic and to compare how the different methods affect the value of these properties.

### 3.5   Juxtaposed lithology

Juxtaposed lithology (a.k.a. Allan diagram) is a representation of the hangingwall and footwall
lithology and their juxtaposition on the fault plane (Allan, 1989; Knipe, 1997). To calculate
juxtaposed lithology (JL), horizons, faults and a well (NO 7120/6-1, Fig. 1b, d) containing
lithological information were used. JL was calculated using the resulting horizon and fault
surfaces from the five experiments. The key lithological units were defined in the well using a
combination of logs, core photographs, information from the NPD Fact pages and post-well
reports. Sonic and density logs were used to generate a well synthetic seismogram, which was
tied to the seismic. Using the same hangingwall and footwall cutoff-lines as in the fault throw
analysis, and the interpreted horizons as guiding surfaces, the well lithologies were projected
onto the faults, and used to generate a JL (Allan) diagram.

### 3.6 Geological modelling and hydrocarbon volume calculations

The geological modelling and volume calculations were conducted on the least and most densely
interpreted experiments (Exps 1 and 4). This analysis was completed using a combination of
structural and property modelling workflows in Petrel™ and the 5 x 5 km study area was



considered to represent the limits of the hydrocarbon field. Firstly, fault and horizon surfaces from Sect. 3.1 were used to create a structural model for each experiment (Fig. 5a). A 3D corner-point grid was generated, and the cells were then populated between the top Fuglen and top

Fruholmen horizons using a grid cell size of 12.5x12.5x1 m (i, j, k direction) matching the resolution of the original horizon surfaces (Fig. 5b). These two horizons define the main reservoir interval (Fig. 5e; Linjordet and Olsen, 1992; Ostanin et al., 2012). In the depth (k) direction, the cells were divided using the proportional method with an approximate thickness of 1 m (~250 cells in total between the top Fuglen and Fruholmen horizons). The grid follows the

shape of the interpreted horizons precisely and the grid pillars align with the fault dip, making an accurate geological representation (Fig. 5b). The faults were included into the grid as zig-zag faults, meaning they were not precisely represented in i and j, but the detailed grid resolution cancelled out most of this effect.  Facies and porosity data (Fig. 5c) were upscaled from the logs of a single well (NO 7120/6-1) to the grid cells at the well locations and then populated across

the structural models for each experiment.  The facies were extrapolated using the sequential indicator simulation method (Fig. 5d). For simplicity, all sands were considered to be net reservoir. A constant oil saturation of 0.9 was used over the whole model for cells located inside the oil-leg.  Finally, an area wide oil-water contact (OWC) was placed at a depth of 2420 m, the deepest point of the top Fuglen surface within the model area, to simulate a spill point with a

footwall trap. Volumes were calculated, including gross rock volume, pore volume and in-place hydrocarbon volume (STOIIP) for both Exps 1 and 4 (Fig. 5f). This simplified modelling was used to quantify the effects of interpretation methodology on the hydrocarbon related volume calculations.

For the volume calculations, there was a concern that any differences between Exps 1 and 4

might be caused, or at least exaggerated, by the stochastic facies and porosity modelling. Different facies and porosity realizations will result in different volumes.  We needed to be certain that any variations in volumetric were caused by the different interpretation methods and not the stochastic property modelling.  Several options were examined to negate this possibility. As the grids are identical in their i, j, k dimensions, it was expected that Petrel™ would produce

the same realization in the 2 grids when the same seed number was selected; this proved to be an incorrect assumption.  The method selected to make sure that the same realizations were being used, and to ensure that an extreme case was not being selected, was to 1) generate 100





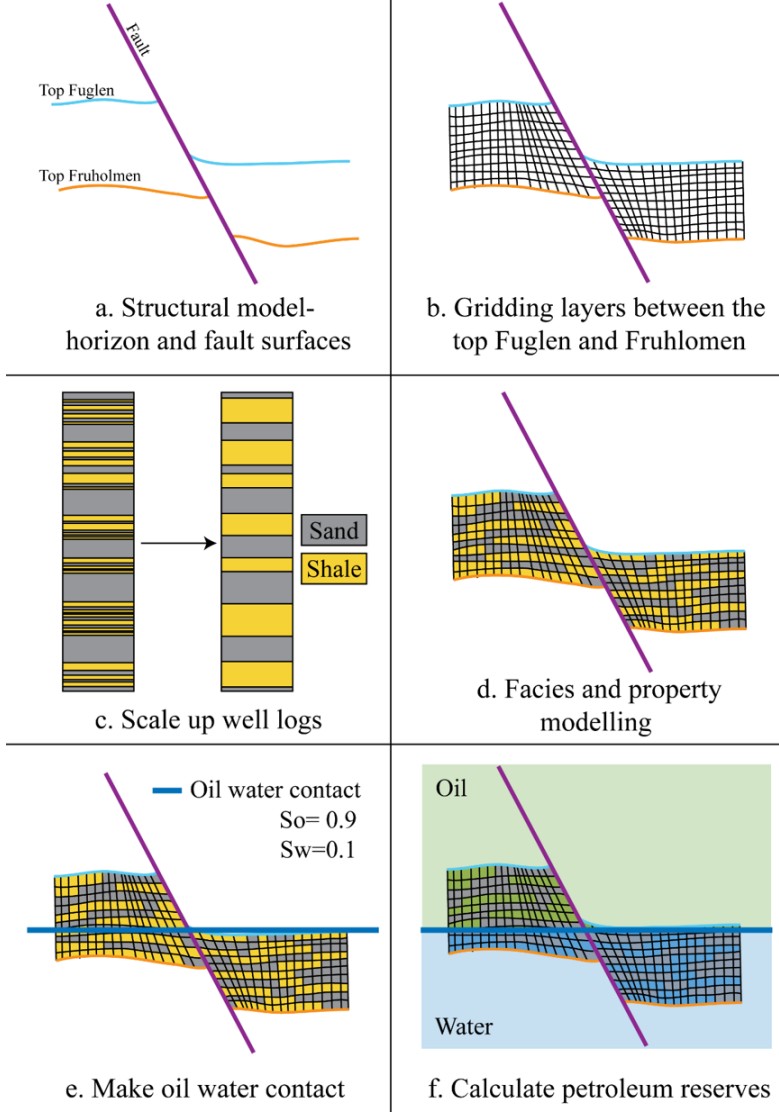

Figure 5: Reservoir modelling and calculation of petroleum volumes method. (a). The creation of the structural model. (b) Establishing gridded layers between the top Fuglen and top Fruholmen. (c) The upscaling of well logs from well 7120/6-1. (d) Populating facies and properties such as porosity into the individual grid cells using the upscaled well log data. (e) Drawing an oil water contact across the study area. This OWC simulates a spill point at the lowest point of the top Fuglen. (f) Running the calculation of petroleum volumes.





realizations on the Exp 1 grid, 2) copy all 100 realizations to Exp 4 grid and 3) then run
volumetric analysis on all realizations for both grids. Once the volumes had been calculated for
100 realizations on each grid, they were analysed to determine the average volumes. This
negated the possibility of selecting an extreme case. Using the same set of realizations in the 2
experiments, meant that the differences in volumes could be assigned, with certainty, to the
differences in interpretation methods used.

**4.  Results**

**4.1 Seismic interpretation**

Five seismic interpretation experiments (Fig. 3) were analysed to understand the effect that the
interpretation methodology has on the resulting fault and horizon surfaces.

Firstly, it is important to consider the areal coverage and visible patterns contained in the
interpretation before it is converted into surfaces (Fig. 3). When analysing the interpretation of
the top Fuglen and top Fruholmen, Exps 1 to 4 have an increase in interpretation density (the
horizon interpretation of Exps 4 and 5 is the same; Fig.3 a-d). All the horizon interpretations
show the same general trends in topography, but as expected the topography is more detailed and
most sharply defined on the most densely interpreted data (Exps 4 and 5; Fig. 3d, e). The top
Fuglen is the most clearly imaged reflector which resulted in complete interpretation coverage in
all experiments (i.e. no gaps in the interpreted lines; Fig. 3). The clear imaging of this reflector is
especially evident in the auto-tracked horizon in Exps 4 and 5 (Fig. 3, top Fuglen). The top
Fruholmen is a poorly imaged reflector which consequently resulted in gaps in the interpreted
lines (Fig. 3, top Fruholmen). The areas lacking interpretation of this reflector are evident in all
experiments but are most clear in the auto-tracked horizon (Fig. 3 d, e; top Fruholmen). The fault
polygons for the two horizons do appear to have the same general trends but this will be
discussed in detail in the next section.

The horizon and fault interpretations were converted into surfaces. The horizon surfaces show
the same general patterns with respect to topography in all the experiments (Fig. 6). Generally,
all top Fruholmen structure maps show a topographic low on the north (hangingwall) side of
each fault. The footwall blocks are uplifted relative to the hangingwalls and the points of highest
elevation are located adjacent to the faults (Fig. 6, top Fruholmen). In the top Fuglen surface, the
same overall topographic patterns are evident, but the amount of footwall uplift and depth of



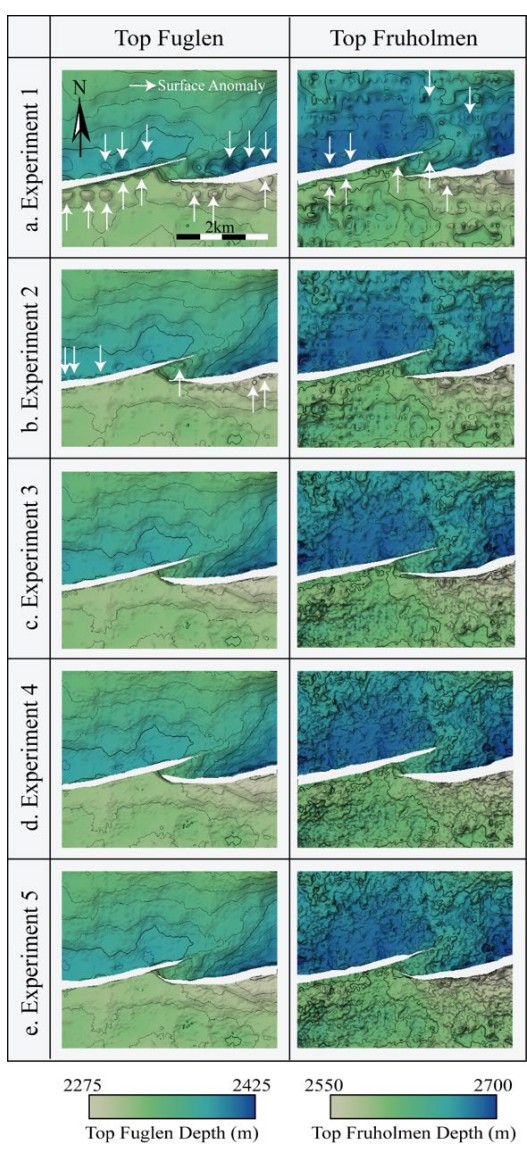

Figure 6: Structure maps of the two interpreted horizons top Fuglen and top Fruholmen (left and right columns respectively). (a) Exp 1 (32 IL x 32 XL interpretation, every 32nd IL faults), (b) Exp 2 (16 x 16, every 16th IL faults), (c)Exp 3 (8 x 8, every 8th IL faults), (d) Exp 4 (3D auto-tracked horizons, every 4th IL faults,). (e) Exp 5 (3D auto-tracked horizons, every 50 m depth slice).





topographic lows on the hangingwall are less than on the top Fruholmen surface (Fig. 6a). The greatest differences between the experiments occur in areas where the lateral continuity of the interpretations were disrupted due to the presence of a fault, where horizon interpretations do not continue across the fault plane and when the interpretation density was low (Exps 1-3; Fig. 6a-c). In these cases, it is possible to identify topographic features near the faults which are clearly

artefacts (Fig. 6a-b; Exps 1-2).

To better visualize the surface anomalies, thickness difference maps were generated between the surfaces of Exps 1-3 and the most densely surfaces of Exps 4-5. Visual inspection indicates that surfaces 1-3 all contain interpretation anomalies. The difference maps show a decrease in thickness difference with increasing interpretation density (Exps 1 to 3). The maps also show

that the top Fuglen surfaces are a closer match to the auto-tracked horizon than the top Fruholmen (Fig. 7).

Exp 1 shows the most significant differences from the 3D auto-tracked horizons due to a sparse interpretation grid and the introduction of gridding anomalies (Fig. 7a). The thickness anomalies in both the top Fuglen and top Fruholmen can measure +/-30 m from the 3D auto-tracked surface

and the anomalous areas are up to 400 m wide and long (i.e. comparable to the interpretation spacing; Fig. 7a). The top Fuglen from Exp 1 correlates moderately well in unfaulted areas and all the major anomalies occur close to the faults (Fig. 7a, top Fuglen). On the hangingwall side of the faults the anomalies are predominantly depressions (i.e. sparse interpretation grid generates a surface that is too deep), while on the footwall side the anomalies trend upward (i.e. the surface

from the sparse grid is too shallow). The top Fruholmen from Exp 1 is more anomalous across the entire surface; there is no clear correlation between the tendencies of the anomalies on the hangingwall and footwall (Fig. 7a, top Fruholmen). The areas of divergence occur at the gaps between interpreted ILs and XLs.

Exp 2 exhibits much less significant changes in thickness with respect to the auto-tracked

horizons on both the top Fuglen and top Fruholmen (Fig. 7b). For the top Fuglen, a pattern like Exp 1 is observed; most thickness anomalies occur near the faults and correspond to gaps in the interpretation (Fig. 7b, top Fuglen). The top Fruholmen is more chaotic, but in this case the anomalies are smaller (up to 200 x 200 m) and exhibit smaller thickness differences (+/-15 m)





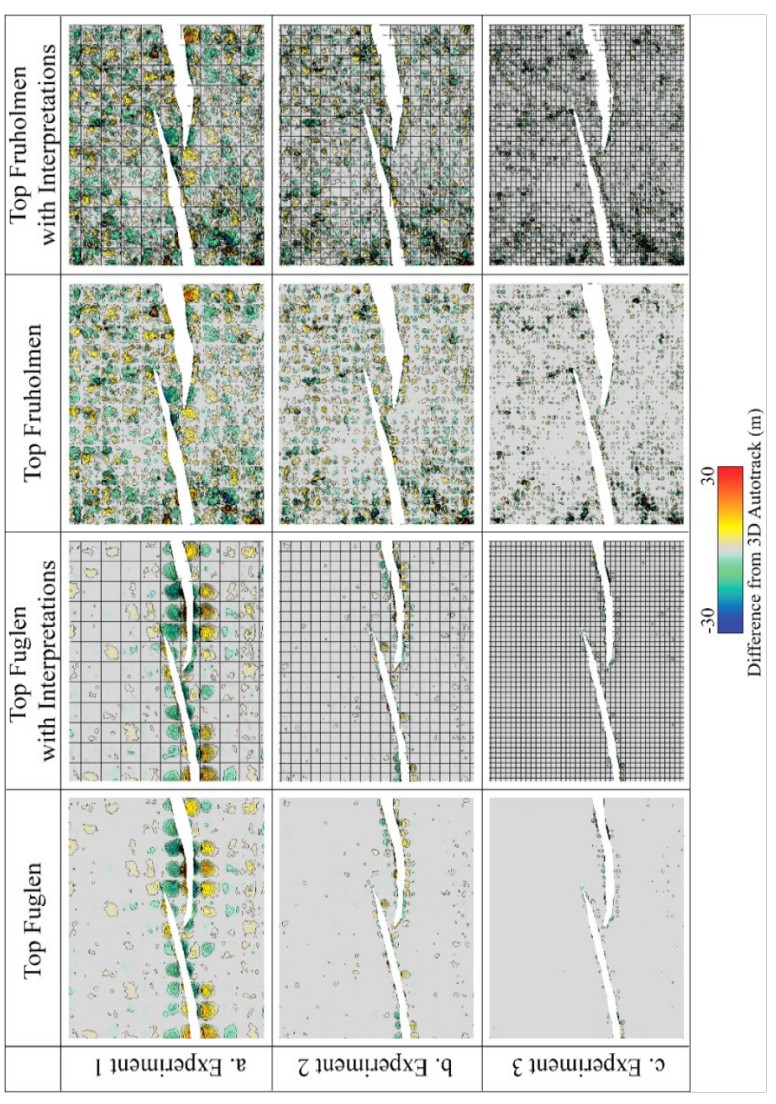

Figure 7: Difference maps of the horizon surfaces for the top Fuglen and top Fruholmen in experiments 1 (a), 2 (b) and 3 (c). The auto-tracked horizon surfaces in Exps 4/5 are the best-case scenario. Difference maps were computed by subtracting the experiments' interpreted horizons from the auto-tracked horizons.

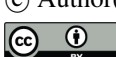



than in Exp 1. Like in Exp 1, the thickness differences in both the top Fuglen and Fruholmen
        correlate with gaps in the interpretation.

        Finally, the thickness anomalies for Exp 3 show the same trends as in Exps 1 and 2, but again
        they are smaller in area (up to 100 x 100 m) and magnitude (+/-5 m; Fig. 7c). The anomalies
        occur at points of gaps in the interpretation. The thickness anomalies in the top Fuglen are almost
always observed near the faults while those on the top Fruholmen are more widespread across
        the whole surface (Fig. 7c). It is important to keep in mind that the top Fuglen has complete areal
        coverage in the study area while the top Fruholmen does not. In Exps 1-3, the thickness
        anomalies in the top Fruholmen structure maps are in some instances linked to inconsistencies in
        the auto-tracked horizon.

**4.2 Fault length and morphology**

        Fault polygons were displayed on structure maps (Fig. 6) and plotted graphically (Fig. 8, 9) to
        show how fault length and morphology changes with the interpretation method. Generally, fault
        length on the interpreted horizons increases with interpretation density from Exp 1 (shortest
        faults) to Exp 4/5 (longest faults). These observations are clear for both the top Fuglen (Fig. 8a,
b) and the top Fruholmen (Fig. 8c, d). In Exp 5 (horizontal fault sticks), the eastern fault is longer
        than the fault interpreted by vertical fault sticks in Exp 4, while the western fault is shorter than
        in Exp 4 (Fig. 8).

        The morphology of the faults also changes with interpretation. In Exp 1 there is a minimal
        amount of interaction between the two very straight faults forming the relay (Fig. 6a). In Exp 2
the faults are also straight and do not appear to interact (Fig. 6b). In experiments 3 to 5 the
        northward curvature and lengthening of the eastern faults towards the western fault increases,
        which suggests that the relay is close to breaching or may even be breached (Fig. 6c-e). This near
        breach relay is evident in the top Fuglen for Exps 4 and 5 but is less prominent in the top
        Fruholmen (Fig. 6d, e).

The effect of interpretation method on fault length is clearly seen in the graphical representation
        of fault trace distance versus fault throw (Fig. 9). The data in these graphs were sampled on the
        interpreted fault sticks and show that in Exp 1 there is minimal overlap between the two faults,
        and the amount of overlap increases towards Exp 4 (Fig. 9a-d). For Exp 5, fault trace distance





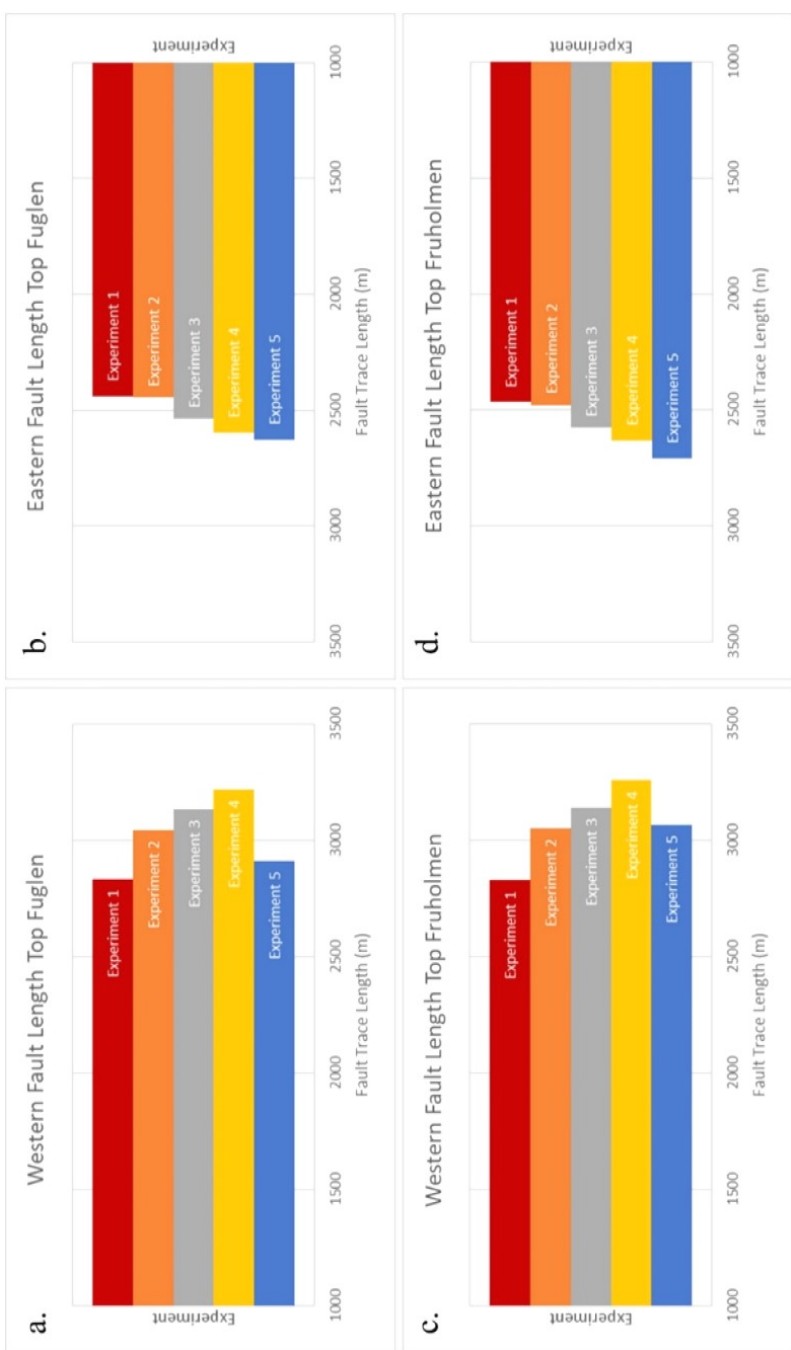

Figure 8: Fault trace lengths of the western and eastern faults for the top Fuglen (a, b respectively) and the top Fruholmen (c, d respectively).






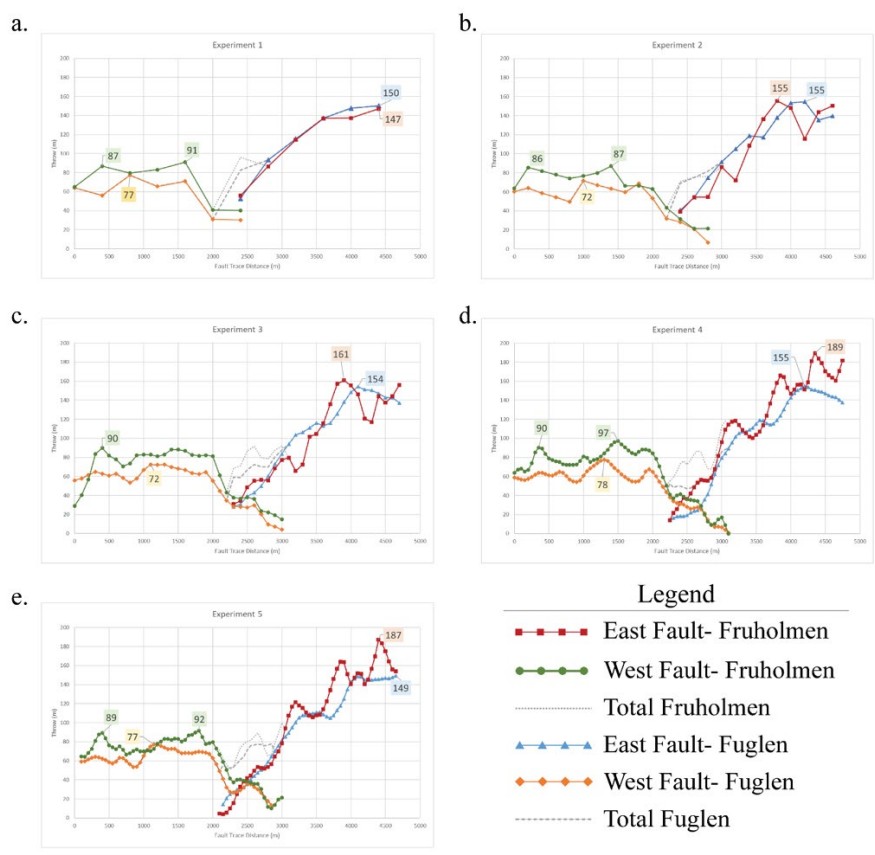

Figure 9: Graphs of fault throw for Exps 1-5 (a-e). Fault throw was extracted for each experiment to match the spacing of the interpreted fault sticks. In Exps 1 (a), 2 (b), 3 (c) and 4 (d), the fault throw was extracted at 400, 200, 100, and 50 m, respectively. In Exp 5 (e), the fault sticks are horizontal. Since it is not possible to extract the fault throw horizontally, the same sampling interval used in Exp 4 (50 m) was used.



versus throw shows that the eastern fault is longer while the western fault is shorter than Exp 4
(Fig. 9e), which confirms our observations from Fig. 8.

### 4.3 Fault throw

Fault throw contours from all five interpretation experiments exhibit in general consistent

patterns (Fig. 4a) on the eastern and western faults, but also some bullseye patterns (Fig. 10a).
The western fault has similar throw magnitudes across all experiments. The lowest throws occur
on the eastern margin and the highest throws (up to 100 m) on the western side. With increasing
interpretation density, the throw results for this fault appear smoother and more laterally
extensive. For example, in Exp 1 the western fault shows three separate bullseye patterns while

in Exps 2 to 4 it shows a progressively smoother throw distribution (Fig. 10a). For the eastern
fault, the throw patterns are similar between experiments, but the throw magnitudes increase
with increasing interpretation density (Fig. 10a). In Exp 1, fault throw reaches a maximum of
~150 m on the eastern side of the fault. For Exps 2 and 3, the results have slightly higher
maximum throw (~175 m) but are segmented into geologically unrealistic bullseye patterns (Fig.

10a). In Exp 4, the maximum throw of the eastern fault is up to 200 m and the results are more
concentric, smoother, and geologically realistic than in Exps 1-3. Exp 5 (fault sticks interpreted
on depth slices) shows similar patterns to those observed in Exp 4 but with more irregularities.

Fault trace distance versus throw also illustrates how fault displacement is influenced by the
interpretation method (Fig. 9). As discussed before, the fault throw of all experiments is greater

on the edges of the study area than near the relay (centre of the graphs in Fig. 9). For the western
fault, the top Fruholmen is always displaced more than the top Fuglen. For the eastern fault, the
top Fruholmen is displaced more than the top Fuglen in Exp 4/5 (Fig. 9d, e) but exhibits similar
throws to the top Fuglen in Exps 1-3 (Fig. 9a-c). In all experiments the throw distributions for
the top Fuglen are smoother than those for the top Fruholmen. This smoothness is also

observable in the throw fault plane projections where the bullseye patterns occur on the top
Fruholmen level. The highest throw values for the eastern fault at the top Fruholmen in Exps 1-5
are ~147, 155, 161, 189 and 187 m respectively. These values occur near the eastern margin of
the study area (Fig. 9). For the western fault, the top Fruholmen peak throw values in Exps 1-5
are ~91, 87, 90, 97, and 92 m respectively. However, these peaks do not always fall near the

western edge of the study area as the western fault is relatively constant in throw outside the





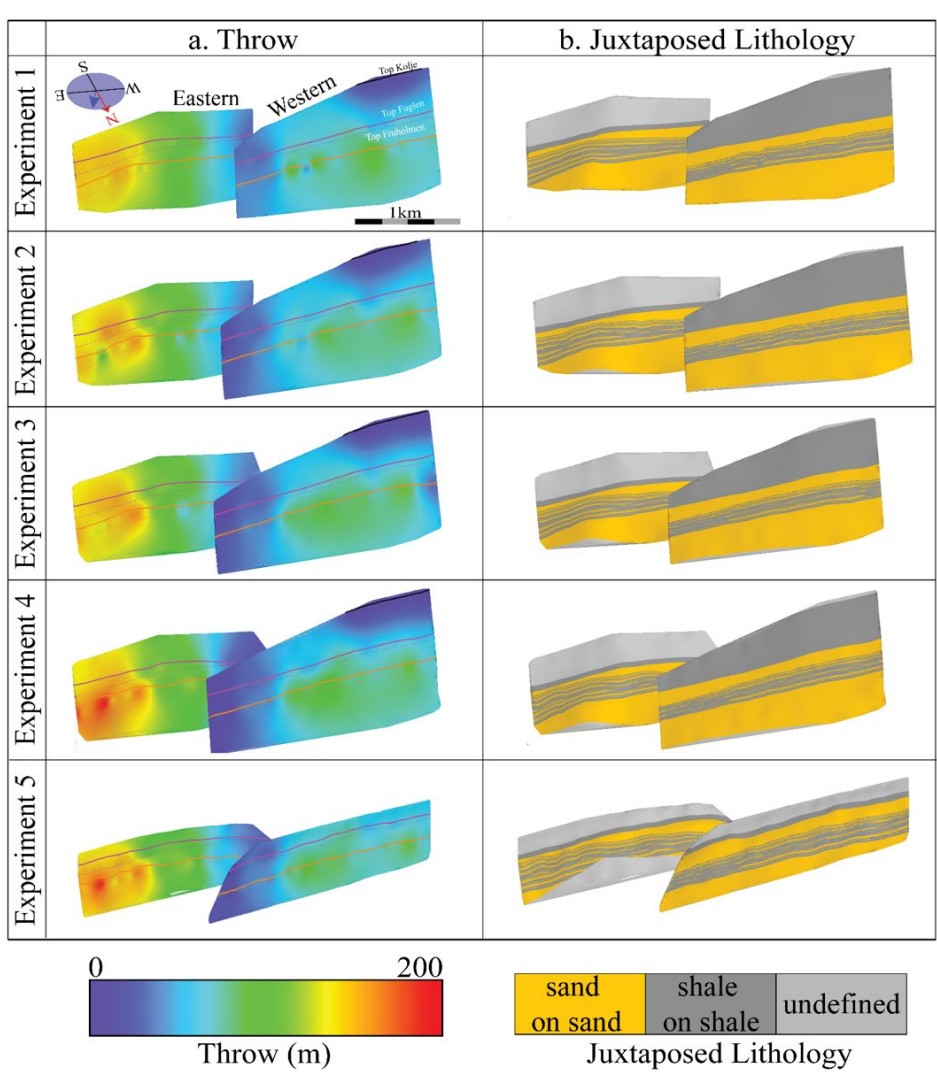

Figure 10: Fault plane projections of (a) fault throw and (b) juxtaposed lithology. The projections are imaged on both the eastern and western faults for all the Exps 1-5.





relay (Fig. 9). The top Fuglen throw on the eastern and western faults has a similar distribution as observed for the top Fruholmen (Fig. 9). At the top Fuglen level, the eastern fault has maximum throws of ~150, 155, 154, 155, 149 m, and the western fault has maximum throws of

~ 77, 72, 72, 78 and 77 m, for Exps 1-5 respectively. Figure 9 clearly shows that the trends of throw for Exp 1 are overly smooth, while those of Exps 2-4 are similar. Exp 5 shows more or less the same result as Exp 4 with slight changes due to the extent of the faults.

## 4.4 Juxtaposed lithology

Lithology data projected on to the fault planes can help us to understand how interpretation

methods can influence the evaluation of reservoir juxtaposition and the potential for fault sealing. All experiments were populated with the same lithological data from well NO 7120/6-1 (Fig. 1b, yellow dot), the only variation is the interpretation method. On a broad scale, the juxtaposition diagrams for the five experiments look very similar on both the eastern and western faults (Fig 10b). The uppermost section of the faults is characterized by shale-shale juxtaposition (dark

grey, western fault) or has not been characterized due to a lack of conformable top Kolje distribution on the eastern side of the study area (light grey, eastern fault). The next unit down is a homogenous sand-sand interval followed by a shale-shale section at the fault centres that is segmented by thin sand-sand units. Finally, the deepest lithology juxtaposition is another homogeneous sand-sand. On closer examination however, comparison of the different

experiments reveal that the lateral extent and definition of the intra-shale sand overlaps improve with increasing interpretation density (Fig. 10b). This is especially true when comparing the least dense seismic interpretation (Exp 1) to the densest (Exp 4). Exp 5 follows the same pattern as Exp 4 in areas where the juxtaposed lithology ran smoothly, but there are some issues with the juxtaposition (light grey triangle at base of eastern fault, Fig. 10b). This anomaly is caused by the

horizontal interpretation of the fault on depth slices resulting in some sections of the fault having vertical dips. It is not possible to generate juxtaposition diagrams in these vertical fault areas.

## 4.5 Dip Slip Gradients (Longitudinal and Shear Strain)

Dip separation gradient (DSG), longitudinal and shear strain (Freeman et al., 2010) were calculated to understand variations in interpretation confidence between the experiments. The

results of dip separation gradient are similar across all five experiments (Fig. 11a). In general, the largest DSG (> 0.2) occur at the top Fruholmen level. The western fault has a larger distribution



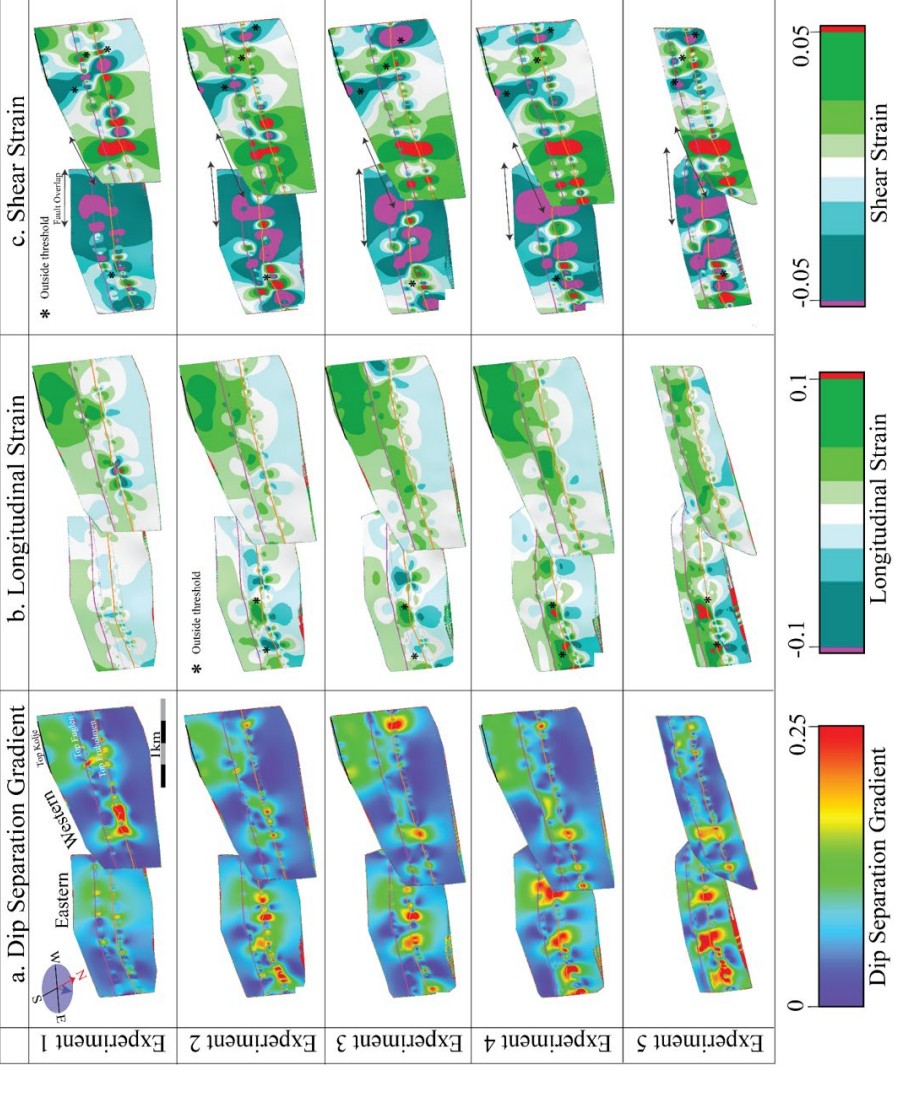

Figure 11: Fault plane projections of (a) dip separation gradient, (b) longitudinal strain and (c) shear strain. The projections are imaged on both the eastern and western faults for all the Exps 1-5.



of high DSG values in the western top part (0.125 gradient), and a main bullseye on the eastern
side (Exps 1, 3-5; Fig. 11a). The eastern fault has the same 3-4 bullseyes occurring in all
experiments, but Exp 1 has the lowest DSG values.

The longitudinal strain (LS) patterns are similar to those observed in the DSG results (Fig. 11b).
The colour bar for longitudinal strain is set so any values outside a geologically realistic
threshold (Freeman et al., 2010) occur as red (LS, > 0.1) or purple (LS < -0.1). The results for LS
for all experiments are similar and exhibit values that are within the defined threshold. In the
western fault for Exp 1, unrealistic LS values at the top Fruholmen level on the eastern side,
suggest a problem with the interpretation (Fig. 10b, top row). This problem is not present in the
other experiments. High (green) LS values in the western upper half of the western fault in Exps
1-4 are within the acceptable threshold (Fig. 10b). These high values coincide with the area
between the top Kolje and top Fuglen. The eastern fault has the same LS bullseyes across its
centre as observed in DSG, but they are mostly within the established threshold. In Exps 4 and 5,
there are two above thresholds high (red) LS areas at the top Fruholmen level (Fig. 11b, black
asterisks). All areas above threshold LS values (red, pink) are less than 250 m across.

For the shear strain (SS), the colour bar is also set to display geologically unrealistic values (+/-
0.05; red and pink, Fig. 10c) (Freeman et al., 2010). Although SS highlights more problematic
areas and places more stringent constraints to the interpretation, it indicates extreme highs and
lows of SS at the overlap of the western and eastern faults respectively (Fig. 10c, black arrows).
The overlapping sections of the fault are more laterally extensive from Exp 1 through to 5, which
is reflected in the lateral extent of extreme SS. Localized (> 250 m) SS bullseyes highlight some
slight interpretation problems discussed before in relation to LS (Fig. 11c, black asterisks). Due
to the high degree of similarity between the experiments no attempt has been made to analyse SS
variations any further.

## 4.6 Reservoir modelling and hydrocarbon volume calculations

In order to test the implications of interpretation techniques on hydrocarbon volume calculations,
the least and most densely populated experiments (Exps 1 and 4) were input through a geological
modelling workflow (Fig. 5). A 5 x 5 km geological model was generated for each experiment
(Fig. 12a, b) and used to calculate the bulk rock volume, pore volume and STOIIP (Fig. 9c, d).



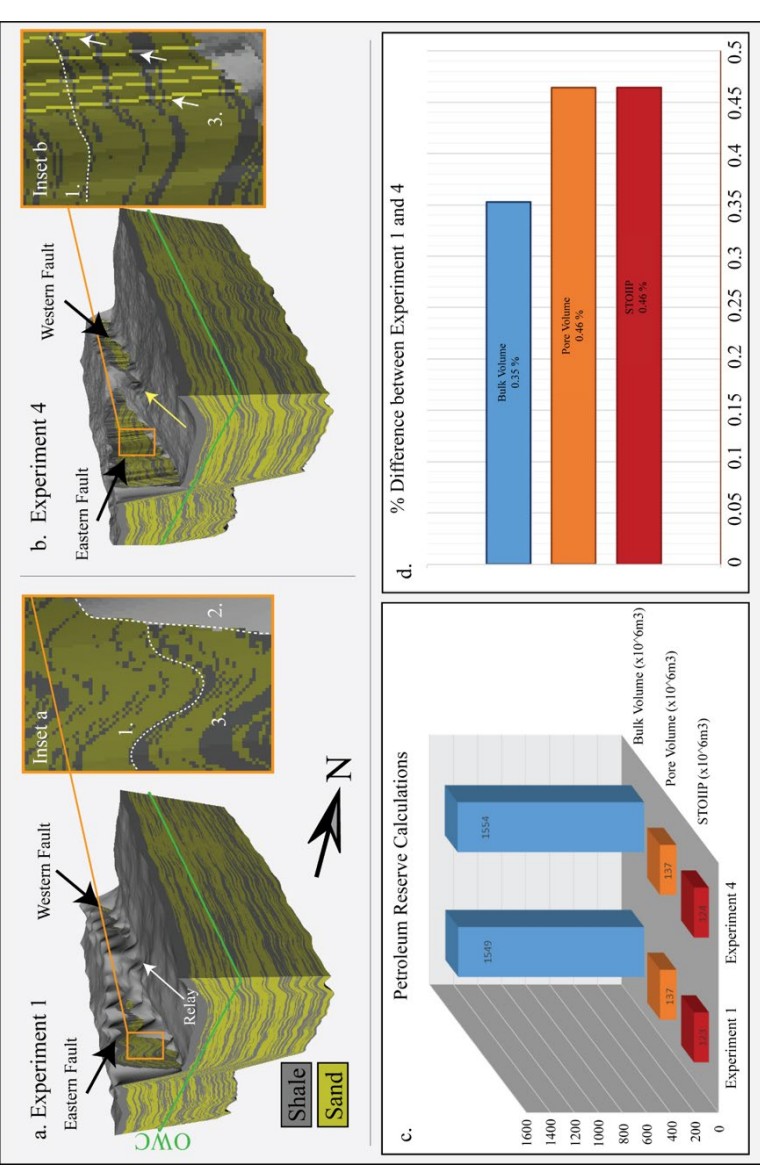

Figure 12: Reservoir modelling and calculation of petroleum volumes. (a) The geological model for Exp 1. (b) The geological model for Exp 4. (c) Graphical representation of the petroleum volume calculations for both experiments. (d) Percent difference of the petroleum volume calculation between the experiments.



There are significant differences in fault morphology, horizon resolution and lithology distribution between the two geological models. In Exp 1, the surface anomalies observed in the structural maps (Sect. 4.1, Fig. 6 arrows) are also evident in the 3D grid at the top and base of the gridded interval (Fig. 12a, Inset a, label 1). Since the top Fuglen and Fruholmen are used as the input to define the top and base of the gridded interval and the cells there within, the surface anomalies also greatly impact the facies distribution in Exp 1, which undulate to match surface anomalies. These facies undulations can be observed on the exposed footwall of the eastern fault and on the eastern geological model boundary as the facies pull upwards towards the footwall (Fig. 12a, Inset a label 1). In Exp 1, there are also some problems with respect to the exposed fault planes where some shale cells have bled up and down the fault planes creating unrealistic peaks (Fig. 12 a, Inset label 2). This results in poor modelling of the relay ramp structure, although the exposed footwall and hangingwall blocks appear relatively smooth (Fig. 12a, Inset a label 3).

In Exp 4, the facies distributions do not have the same undulations that are observed in Exp 1. This result is more or less expected since these anomalies were not evident in the top Fuglen and Fruholmen which define the grid. Flat, more geologically representative facies distributions are clear on the uplifted footwall of the eastern and western faults and on the exposed eastern boundary of the model (Fig. 12b, Inset b label 1). A 'bleeding of facies' occurs on the margins of the model and slightly on the edges of the faults (Fig. 12b). The relay ramp is much more clearly defined in this experiment than in Exp 1 (Fig. 12b, yellow arrow). The faults are better defined with respect to length and morphology, but the high density of interpreted fault sticks means that the fault planes have vertical jumps between grid cells in the 3D grid (Fig. 12b, Inset b label 3).

Bulk rock volume, pore volume and oil (STOIIP) were calculated for both geological models, using an oil water contact of 2420 m (Fig 12a, b; OWC). This contact was chosen to mimic a spill point at the lowest point on the top Fuglen surface. The volumetric analysis was run on each of the 100 realizations, the results presented are given as their average. The stochastic facies and porosity realizations used in these calculations were identical for the 2 experiments, which allowed any volume differences to be assigned to the impact of the resolution of the interpretation. The volumetric calculations for Exp 4 were always slightly larger than Exp 1. The bulk volumes for Exps 1 and 4 are 1548.7 and 1554.2 x $10^6$ m$^3$ respectively (a difference of



0.36%). For pore volume 136.8 and 137.4 x10$^6$ m$^3$ were calculated from Exps 1 and 4 respectively, which is a difference of 0.46%. Finally, the calculation of oil in place (STOIIP)

resulted in 123.1 x10$^6$ m$^3$ for Exp 1 and 123.7 x10$^6$ m$^3$ for Exp 4 (a % difference of 0.46%).

The volumes in Exp 4 are slightly larger than in Exp 1, with the increase in the bulk rock volume carried through the pore-volume and STOIIP calculations. However, the percentage differences are very small, less than 0.5% for all metrics.

## 5. Discussion

### 570 5.1 Implications on horizons and faults morphologies

The seismic interpretation method had a significant impact on all aspects of the fault analysis workflow. We found that both Exps 4 and 5 provided the most geologically accurate representation of the morphologies of horizons, faults, and their intersections. The eastern fault was longest in Exp 5, while the western was longest in Exp 4 which suggests a combination of

the methods (i.e. vertical and horizontal interpretation) would be the most rigorous approach to fault interpretation. The horizons in Exps 4 and 5 were quick to interpret because of 3D auto tracking and were the most detailed. When interpreting the Top Fuglen there was no need for a QC process since the imaging of this reflector was clear and the final surface did not contain any artefacts in the interpretation (Fig. 7, Top Fuglen columns). The Top Fruholmen needed some

manual guidance/QC and did have some interpretation artefacts but this was unavoidable due to the poor seismic quality (Fig. 7, Top Fruholmen columns). The interpretation of faults was slightly more time consuming for Exp 5 relative to 4 but the attribute volume increased the understanding of fault morphology and length compared to Exp 4 (see Fig. 8, fault lengths). Exp 1 is here considered to be a failure with respect to observed geological morphologies and this

methodology cannot be recommended as a method for fault interpretation, even though it was very time efficient. The sparsity of the horizons and fault interpretations led to inaccuracies and gridding anomalies proportional to the spacing of the interpreted inlines (400 m), reduced fault length (Fig. 8, up to 400 m difference between Exps 1 and 4, western fault), and incomplete understanding of the relay morphology. Exps 2 and 3 were an improvement on Exp 1, as

expected. They captured some important information but not as much as Exp 4/5. The differences between Exps 2 and 3 were much less significant than those between Exps 1 and 2.



As such, if manual interpretation of faults is required then Exp 2 should be considered as the minimum acceptable interpretation density for performing a detailed fault analysis workflow.

The two aspects of the fault analysis workflow that were the most effected by the interpretation method were fault length and throw. Both the length and throw of the faults differed dramatically depending on interpretation density which in turn had a large influence on the apparent morphologies of the faults and of the relay ramp (Figs 8-10). The knock-on effects of these are because the fault lengths and throws impact all other aspects of the workflow. Overall, comparison of the most and least densely interpreted datasets (Exps 4/5, 1 respectively) show that the length, morphology and throws were different at both the Top Fuglen and Fruholmen level (Figs 7-10).

The impact of the interpretation method on the length, morphology, and throw profiles in the relay is critical to understand its formation. Fault displacement/throw relationships in relay ramps are dependent on the stage of relay development in question (Fig. 13). In the first stage of relay development, the faults do not overlap and therefore exhibit isolated fault throw profiles (Barnett et al., 1987; Fig. 13 a-b). Stage 2 of relay development is defined by the propagation of faults to form a relay ramp (Fig. 13c). Fractures break up the ramp (that in our case are sub-seismic resolution) and accommodate some of the strain of the relay (Larsen, 1988; Peacock and Sanderson, 1994). The throw profiles of the faults interact and the total throw of the overlapping fault segments is accommodated by the relay ramp (Peacock and Sanderson, 1994; Fig 13d). The fault extents and throw profiles for Exp 1 (Fig. 9a, 10a) fall somewhere between stage 1 and 2, where there is a slight overlap of the faults, but a relay is only just starting to form (Fig. 6a). This is because Exp 1 does not properly capture the full length of the fault. Stage 3 of relay development is defined as when the faults have continued to propagate and fractures have begun to spread through the relay structure as it is near the maximum amount of strain it can accommodate (Long and Imber, 2012; Peacock and Sanderson, 1994). The propagation of the fault tips toward the relay and increased fault overlap are evident (Fig. 13e-f). Stage 4 of relay development defines the destruction (breaching) of the relay ramp and the formation of branch lines between the two relay forming faults (Peacock and Sanderson, 1994). The original tiplines of the fault are no longer active, and the faults are now joined along branch-lines formed in the weakened and sheared ramp margins (Fig 13g-h). When analysing Exp 4/5, the morphologies are



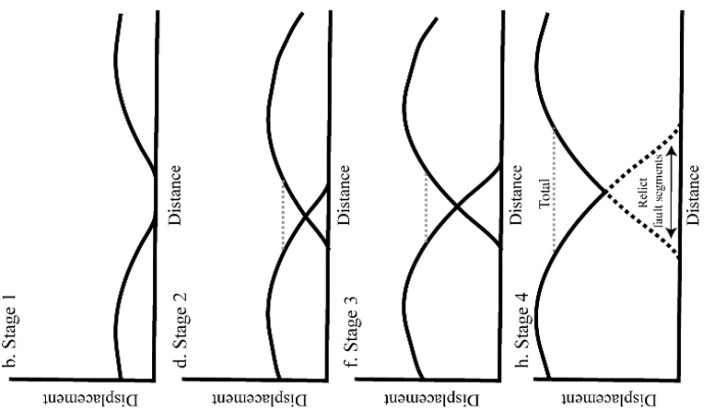

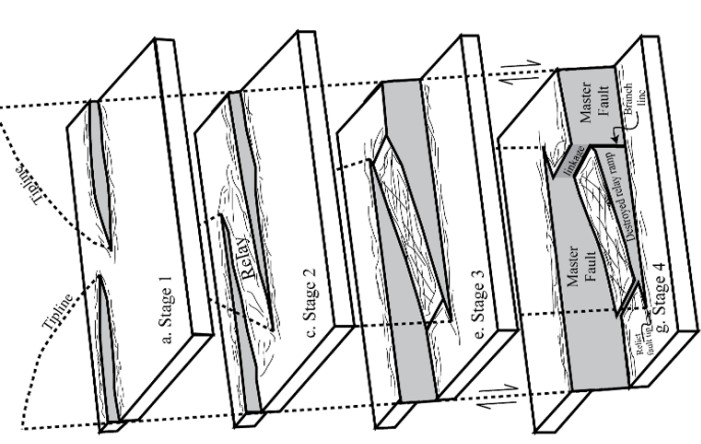

Figure 13: The stages of the relay and the displacement distribution of those stages. Stage 1 (a, b), Stage 2 (c, d), Stage 3 (e, f) Stage 4 (g, h). The displacement of the isolated faults in stage 1 follows Barnett et al. (1987). Figures modified from Fachri et al. (2013a), Long and Imber (2012), Peacock and Sanderson (1994) and Rotevatn et al. (2007).





comparable to those observed in stage 3 of the relay formation. The northward propagation and
curvature of the eastern faults tipline is clear, and there are likely fractures forming in the relay
that are below the resolution of the seismic data. The relay in Exp 4/5 has not breached on either
the Top Fuglen or Fruholmen level, although it is very close to breaching in Exp 5 at the Top
Fuglen (Fig. 6d/e, 9d/e, 10a). The potential impact of a relay on a working hydrocarbon system
and the implications of misinterpreting the relay are discussed in the upcoming Sect. 5.2.

A study of longitudinal and shear strain was completed to test the accuracy of the interpretation
methods (Freeman et al., 2010). According to Freeman et al. (2010) longitudinal and shear strain
values in isolated faults should remain inside their defined threshold values (+/- 0.1 and +/-0.05
respectively) in order for the interpretation to be deemed accurate. High and low values of
longitudinal and shear strain were observed across all experiments, some of which are outside
these defined thresholds (Fig. 11 b, c). There is a high and low shear strain accumulation in all
experiments on the western and eastern faults respectively in the parts of the faults exhibiting
overlap (Fig. 11 c). Freeman et al. (2010) stated that in the event of overlapping faults, higher
shear strains (above their defined limit) are to be expected in the overlapping segments of the
fault. The limits in this case are higher than could be expected from an isolated fault (Freeman et
al., 2010). These highs and lows appear to change with interpretation density and align with the
increased overlapping of the faults (Fig. 11 c, double ended black arrows). There were some
bullseye patterns (longitudinal and shear strain plots) which were outside of the fault overlap and
outside of the defined threshold strains, these are interpreted to be artefacts produced by
incorrect fault stick interpretations (Fig. 11 b, c black asterisks). It is important to note that
interpretation accuracy with respect to longitudinal strain and shear strain was not the aim when
running the initial interpretations, and therefore it is expected that some inconsistencies are
present.

## 5.2 Implications on petroleum studies

### 5.2.1   Interpretation and aspects of the petroleum industry

Relay ramps and the faults that define them have significant impact on sediment distribution
pathways (deposition of reservoirs), fluid flow/ migration pathways, fault seal/juxtaposition and
trap definition (e.g. Athmer et al., 2010; Athmer and Luthi, 2011; Botter et al., 2017; Fachri et
al., 2013a; Knipe, 1997; Manzocchi et al., 2008a, 2010). By under interpreting the relay with




respect to fault length and throw (as discussed in Sect. 5.1, Exp 1) there is a clear misunderstanding of the stage of relay development, and therefore a misunderstanding of fault interactions. Exp 1 exhibits shorter faults with less throw and therefore a less defined relay (Fig. 14, left column). This under-interpretation of the relay will also have implications on our understanding of sediment distribution pathways (Fig. 14a). Compared with the relay interpreted from Exp 4 (Fig. 14, right column), the results of Exp 1 (left column) also show: less laterally continuous extent of juxtaposed sand-on-sand resulting in different fault sealing (14b), an unsuccessful fluid flow schematic where petroleum does not migrate towards the producer well (14c), and an under estimation of trap size because of the incorrect trap geometry (Fig. 14d). These results are specific for our field area / relay morphology and of course, may differ with changing field parameters. The important thing however is that significant differences can be generated by applying an interpretation method that is unsuitable for the scale of the structures that are being analysed.

### 5.2.2   The effect of interpretation on geological modelling

A geological modelling workflow was run on the least and most successful interpretation methods (Exps 1 and 4 respectively) in order to understand the impact of the interpretation method on the geological model. In Exp 1 it is possible to identify several clear inaccuracies and problems with the model. The problems include facies undulations which were caused by interpretation sparsity, facies bleeding on the fault planes, and the apparent under interpretation and imaging of the relay ramp due to under interpreted faults. The observed facies undulations can have significant implications if used in dynamic modelling processes such as fluid flow simulations. Since the relay is so under-interpreted in Exp 1 the results can be expected to be false. This poor interpretation can have negative implications for the development of the field, production strategies, drainage strategies and may influence the complete field understanding.

The bleeding of facies on the fault planes is caused by the low interpretation density and is easily avoided with a denser interpretation. Exp. 4 had more realistic horizon morphologies, more geologically realistic facies distributions and much less facies bleed. The only problem with this interpretation was that the inline fault stick spacing resulted in linear cell anomalies and unsmooth fault planes (Fig. 12b).  Therefore, we suggest that when modelling, the removal of fault sticks in the fault's centre may provide clearer results.


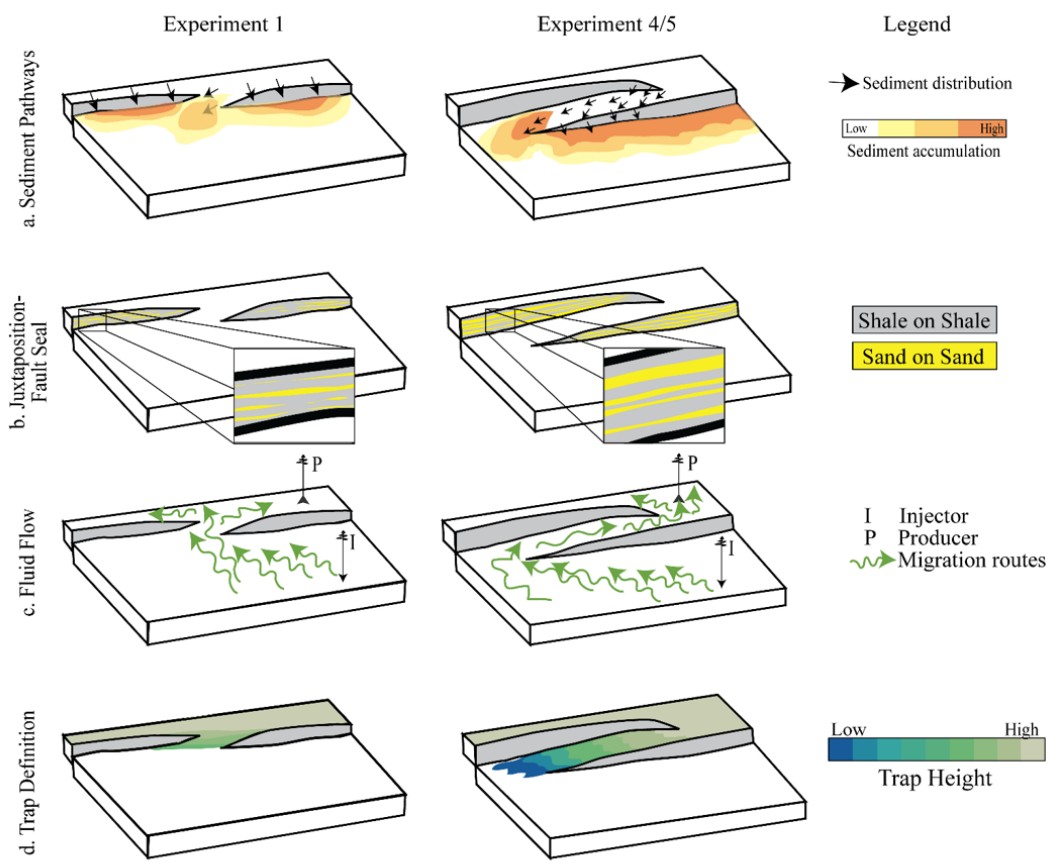

Figure 14: A The comparison of sediment distribution pathways (a), lithological juxtaposition/ fault seal (b), fluid flow (c) and trap definition (d) on an under interpreted version of a relay (Exp 1, column 1) and an accurate interpretation of the relay ramp (Exp 4, column 2). Figures based on Athmer et al. (2010), Athmer and Luthi (2011), Botter et al. (2017), Fachri et al. (2013a), Knipe (1997), Peacock and Sanderson (1994) and Rotevatn et al. (2007).



Volumetric calculations using the two models revealed that the gross rock volumes were 0.35%
larger in Exp 4 when compared to Exp 1 and both the in-place hydrocarbon volume (STOIIP)
and pore volume calculations of Exp 4 were 0.46% greater than Exp 1. These differences are
small (and certainly much less than the normal uncertainty values considered in the industry)
which does suggest for preliminary field analysis/ petroleum calculations, a detailed seismic
interpretation is not all that important. However, this result has significant implications when
upscaled to the fields dimensions – in this case the Snøhvit field in its entirety. For simplicity in
the calculations we take the values from the Norwegian Petroleum Directorate for field size and
the STOIIP in the entire Snøhvit area to be referencing an oil only field. In reality, the field
contains gas, condensate and a small oil column (NPD, 2020). According to the Norwegian
Petroleum Directorate, the Snøhvit field holds in place volumes of ~400 x10$^6$ m$^3$ oil equivalent
(NPD, 2020). A STOIIP difference of 0.46% between Exps 1 and 4 on this field size is equal to
~1.84 x10$^6$ m$^3$ oil in place. This is equal to an underestimation of ~11.6 million barrels (1 m$^3$ oil
= 6.29 bls) of in place oil in Exp 1 versus 4. The NPD lists the recovery factor of the Snøhvit
field to be 64% (NPD, 2020) so only 7.4 million barrels can be considered recoverable.
Assuming an oil price of 50 USD per barrel, this difference in interpretation method is equivalent
to c. 370 million USD. Although this value is relatively small in the industry, it is staggering to
consider how the inaccuracy in the calculation of petroleum reserves can be solely based on poor
interpretation strategies which are mistakes that are completely avoidable.

## 5.3 Recommendations for best practice seismic interpretation

### 5.3.1 Horizons and horizon-fault intersections

The results showed that 3D auto tracking (1 x 1 density) gave the best results in terms of detail in
the structure of horizons, horizon-fault intersections (cutoff-lines, throw, etc.) and was the most
time efficient option assuming relatively high-quality data. In the case of high-quality data and
well-defined continuous strong seismic reflectors (e.g. Top Fuglen), little manual quality control
of the interpretation is required. If the seismic data is of poorer quality, or the reflector in
question is poorly imaged, discontinuous or changes seismic polarity, or there is significant
structural complexity and ambiguity, then it is important to reflect on the task at hand. This is
because auto tracking algorithms may fail or generate artefacts or erroneous results that require
significant manual adjustment to correct. If fault seal or juxtaposed lithologies are critical to the





field analysis, then a denser manual/ 2D auto-tracked method might be necessary and worth the

715 significant time commitment (i.e. 8 x 8). If detailed structural analysis is not required then a less

dense (i.e. 16 x 16) grid will give sufficient results for geological interpretation, while the results

of this study showed that a sparse interpretation spacing (i.e. 32 x 32) gave a geologically

unrealistic and inaccurate representation of the subsurface that could lead to critical errors in

prospect or field evaluations and, as such cannot be recommended except for broad-scale

regional understanding. These results assume a 12.5 m IL and XL spacing and may need to be

adjusted in the event of a different spacing.

### 5.3.2 Faults

The results of Exps 4 and 5 are very similar and give the most accurate results with respect to

fault extent, throw and morphology of the relay. When considering our experiments, it was

725 difficult to capture the entire fault length if using less than a 4 IL spacing, but we also found

interpretation on horizontal time/depth slices to be a useful tool to accurately capture fault length

in its entirety. Therefore, the recommendations are to interpret faults on a minimum of 8 or 16 IL

spacing for the main body of the fault, and on approaching tip lines or complex fault

intersections to decrease the line spacing in order to capture the full length, morphologies and

730 relationships. We also recommend the combination of horizontal fault sticks and attributes to

understand fault morphology, fault extent and to keep track of fault locations in 3D when

interpreting horizons. The results shown here demonstrate that less than 16 IL spacing was

insufficient to capture critical details required when performing fault interpretations, and as such

should be avoided for critical prospect or field scale mapping. These results are also assuming an

735 IL/XL spacing of 12.5 m and may need adjustment if the data differ.

### 6. Conclusions

This paper has analysed the effect of the seismic interpretation method on faults, horizons and

their intersections and shows some of the implications of these interpretations on the results of a

fault analysis workflow. The main findings are summarized as follows:

• Interpretation: The density of fault and horizon interpretations are critical to understand

fault relationships and morphologies in structural maps. The 3D auto-tracked horizons

and a combination of vertical and horizontal fault sticks give the best results in the





relatively high-quality Snøhvit seismic data with moderate to very clear continuous
seismic reflectors. However, in other areas or on poorer data, a combination of auto
tracking or dense 2D interpretation grids are required to properly capture the geological
complexity.

- Fault length is greatly impacted by the interpretation method. Special attention and
denser interpretation are needed around fault tiplines and the least dense experiments did
not capture these extents.

- The biggest effect on fault throw (and therefore much of the fault analysis workflow) was
the interpretation density. If fault seal or dynamic simulation is critical, then denser
vertical sticks (8-4 IL) give the most accurate morphology of faults, despite needing more
time and manual QC.

- Longitudinal and shear strain are excellent tools for understanding interpretation accuracy
and were proven higher in the relay (as observed in Freeman et al., 2010). Studies of
complex faulted fields and prospects should consider implementing these methods if
robust fault interpretation is critical for geological understanding.

- The example showing the effect of interpretation method on geological modelling and the
subsequent calculation of petroleum reserves showed that the importance of correct
interpretation should not be underestimated. The most geologically realistic results were
established when using the densest interpretation (Exp 4). If using Exp 1 interpretations
as the model, the results were less geologically accurate (undulating facies, creeping fault
cells) and resulted in an under-interpretation of the relay, all of which has implications
for dynamic modelling techniques such as fluid flow simulations, production and
drainage strategies.

- Calculations of petroleum reserves in the modelling resulted in an under estimation of
STOIIP of 0.46% when comparing Exp 1 to 4. The upscaling of this value across the
Snøhvit field results in an under estimation of ~11.6 million barrels or ~370 million USD
when comparing Exp 1 to 4. Although this seems small on industry standard, this
difference is only caused by inaccuracy of the seismic interpretation method. These
inaccuracies in modelling and subsequent economic analyses could be almost completely
avoided by applying more robust interpretation methods.





**Author Contributions**

Jennifer Cunningham designed and interpreted the five interpretation experiments in this paper with scientific contribution from Nestor Cardozo. Both Jennifer and Nestor collaborated on the creation of the fault analysis workflow while the application of the workflow on the five experiments was completed by Jennifer. Chris Townsend and Jennifer collaborated on the design of the geomodelling aspect of the workflow and its final implementation and calculated

petroleum volume estimations in the study area. Richard Callow aided in the implementation, and upscaling of the petroleum reserve calculations and contributed on discussions related to the petroleum implications. Jennifer drafted the manuscript and figures with scientific contributions and proofing from all co-authors.

**Acknowledgements**

The authors would like to thank the Norwegian Ministry of Education and Research for funding this research. Thanks to Equinor ASA and their partners in the Snøhvit field, Petoro AS, Total E&P Norge AS, Neptune Energy AS, and Wintershall DEA AS for providing the seismic data used in this study and technical guidance when analysing the seismic data. We would also like to

thank Petrel™ and Badleys (T7™) for providing us with the licenses for their softwares and their support.

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
