# Peer review of "The Impact of Seismic Interpretation Methods on the Analysis of Faults: A Case Study from the Snøhvit Field, Barents Sea"

_Solid Earth, 2020_

## Referee Comment (RC1) · David Tanner (Referee) · 17 Nov 2020

Dear authors, This a good manuscript that deals with a novel and valid idea; how much detail is required when modelling faults and horizons from 3D seismics. How does one quantify the work involved to get the amount of detail needed to continue into petroleum analysis? I think the whole paper is well written. I have only minor comments, see below.

Yours sincerely, David Tanner

- Does the paper address relevant scientific questions within the scope of SE? *Yes*

[Figure]

- Does the paper present novel concepts, ideas, tools, or data? *The idea is novel, the data and tools are not.*

- Are substantial conclusions reached? *Yes*

- Are the scientific methods and assumptions valid and clearly outlined? *Yes*

- Are the results sufficient to support the interpretations and conclusions? *Yes*

- Is the description of experiments and calculations sufficiently complete and precise to allow their reproduction by fellow scientists (traceability of results)? *Yes*

- Do the authors give proper credit to related work and clearly indicate their own new/original contribution? *Mostly*

- Does the title clearly reflect the contents of the paper? *Yes*

- Does the abstract provide a concise and complete summary? *Yes*

- Is the overall presentation well structured and clear? *Yes*

- Is the language fluent and precise? *Mostly, see below*

- Are mathematical formulae, symbols, abbreviations, and units correctly defined and used? *Yes*

- Should any parts of the paper (text, formulae, figures, tables) be clarified, reduced, combined, or eliminated? *No*

- Are the number and quality of references appropriate? *Yes*

- Is the amount and quality of supplementary material appropriate *NA*

General points:

- (lines 65–78) This part of the introduction needs more workflow references. I would suggest the Fault Analysis Group, Basin Analysis Groups, etc. I do not like to see so much self-citation, especially in the introduction, as can be seen in this part. Please revise.

- (line 174) You use the term 'manual horizon interpretation techniques' to include 2D auto-tracking. Please write a sentence on the parameters of the 2D auto-tracking.

- (lines 210-213) What seeding did you use for the 3D-tracking algorithm?

- (lines 165–355) Please transfer any results from the methodology to the results. For instance, time required for the various experiments and methods could be the first part of the results.

- (lines 640–643) You interpret the bulleye patterns as artifacts. This need not be so; they could be real local maxima that occurred due to the coalescence of original faults. Please expand this section, for instance, see Lohr et al. Evolution of a fault surface from 3D attribute analysis and displacement measurements — J. Struct. Geol. 2008, doi: 10.1016/jsg.2008.02.099

- (lines 678–683) This might be throwing the baby out with the bathwater! Removing the fault sticks in the middle of the fault might make dynamic modelling of the fault less difficult, but maybe the undulations on the fault are real. Faults may have a large degree of corrugations (in the slip direction), which has been proven to be greater in scale than the seismic resolution (eg. Needham et al. (1996), Resor & Meer (2009), Ziesch et al. (2017)– doi: 10.1111/bre.12146)

- English. I added about 50 commas as I read the manuscript. Commas make the text flow better. Use a comma before 'respectively', 'which', for instance. On the subject of language; please stick to American English, if that's what you

prefer. Be careful of tense in, for instance the introduction [line 97] "Our study will test.."- Our study tests.. [line 100] "We have designed" - We designed. Maybe you should get a native speaker to read the final manuscript.

Minor points:

- (Fig 1) Highlight top Fuglen and top Fruholmen in the stratigraphic column.

- (Fig 2) Add 1, 2, 3, 4 above columns. Then please rewrite the caption.

- (Fig 9) This figure is hard to read as set in the draft manuscript. What are numbers in the graphs? Local maxima?

---

## Referee Comment (RC2) · Graham Yielding (Referee) · 7 Dec 2020

This manuscript presents the results of a seismic interpretation experiment where the line density of interpretation is systematically varied. The results are tested by measuring various structural geological attributes. As the authors imply, the conclusion is intuitively obvious (denser interpretation = better), but surprisingly no work has previously been published to this effect (though the idea has been employed in training courses by me, my colleagues, and others across the petroleum industry). Therefore it is well worth publishing, with only minor revision.

There are a number of very relevant references that are not quoted, and which ap-

proach the same problem from a slightly different angle. Several deal explicitly with the relay ramp geometry, or more generally with the issue of fault linkage in sparse data. For the benefit of the authors they are:

* Solum et al 2016. "Increasing interpreter capability in structurally complex settings....". In: 3D Structural Interpretation: Earth, Mind & Machine, AAPG Mem. 111.

* Yielding & Freeman 2016. "3-D seismic-structural workflows....". Also in AAPG Mem. 111.

* Richards et al 2015. "Interpretational variability of structural traps....". In: Industrial Structural Geology, Geol.Soc. SP 421.

* Wood et al 2015. "The missing complexity in seismically imaged normal faults....". Also in SP 421.

* Needham et al 1996. "Analysis of fault geometry and displacement patterns." In: Modern developments in structural interpretation...". Geol.Soc. SP 99.

Minor comments for the authors:

* The abbreviations IL and XL are used in two senses - inline and crossline, as initially defined at line 38, but also "inline spacing" and "crossline spacing", for example in Fig.2. The text in Figure 2 might be read as meaning 32 inlines were picked in Expt.1, 16 inlines in Expt.2, etc. Keep IL and XL for just inline and crossline, and add "spacing"wherever it is needed.

* line 180: Explain that "2D auto-tracking" means tracking along a horizon on a vertical section. Additionally explain the exact technique used - I assume you picked endpoints of each horizon segment in each fault block, rather than picking the two ends of the section for the auto-tracker to work along the entire faulted profile. The issue of what an autotracker does at fault breaks is very significant in terms of the time needed for remedial editing, particularly moving to the 3D case. In my experience there will always be cases where the tracker can erroneously go across the fault onto a different

reflection - see discussions in Yielding & Freeman (2016) and Needham et al (1996), listed above. The best way to stop this is to pick the fault planes first, then use them as lateral boundaries for the autotracking.

* line 230 et seq: I recommend you include an example of a tensor slice (in or after Fig.3) to illustrate this form of data. I assume it looks like a coherence slice?

* line 284: reference here to Top Kolje - you should add this horizon to Figure 1d. so that the reader can see where it is on the seismic.

* Figure 4b - you might want to borrow a map+3D version of this from figure 10 of Yielding & Freeman 2016.

* line 496 - mention that this issue is a software limitation, not something inherent in the structural geology.

* Section 5.2.2: you have concentrated on the volumetric implications in your discussion. I think you should maybe add a few sentences to describe the implications there might be for reservoir development. Subtleties in fault linkages, as revealed by better interpretation, might have a big effect on fluid flow during production (e.g. Fig.13 of Jolley et al 2007, Petroleum Geoscience, 321-340).

Some typos....

* As mentioned by David Tanner, there are lots of missing commas, which impedes the fluency of reading. Ensure that subordinate phrases and clauses having a closing comma.

* line 106 - should be e.g. not i.e.

* line 148 - the old software name was TrapTester with both T's upper-case.

* line 162 - orientation not orientations.

* line 248 - horizon not horizons.

* line 501 - occurs not occur.

---

## Referee Comment (RC3) · Graham Yielding (Referee) · 7 Dec 2020

The link to the Badley et al 1990 reference is broken. Use this instead (1991):

http://archives.datapages.com/data/specpubs/resmi1/data/a164/a164/0001/0200/0224.htm

---

## Short Comment (SC1) · 12 Dec 2020

Hi David,

Thank you so much for reading and reviewing our manuscript. We really appreciate the attention you have paid and your time. I have begun implementing these changes.

Enjoy the holidays, and again thank you for taking the time.

Jennifer

---

## Short Comment (SC2) · 12 Dec 2020

Hello Graham,

Thank for your the extensive list of articles. This is exactly why we needed a fault expert like you to contribute to this manuscript. The missing citations and adjustments to the manuscipt are now in the process of being implemented.

We appreciate you taking the time to read our paper and look forward to working with you again in the future.

Best regards and best wishes this holiday season.

[Figure]

Jennifer

---

## Author Comment (AC1) · 5 Jan 2021

Hi David, Thank you so much for reading and reviewing our manuscript. We have now implemented the suggested changes.

Jennifer Cunningham
* * *

---

## Author Comment (AC2) · 5 Jan 2021

Thanks again Graham. The comments were incredibly helpful. The suggested changes have been implemented as you requested.
* * *

---

## Author Comment (AC3) · 5 Jan 2021

Thanks again Graham. The comments were incredibly helpful. The suggested changes have been implemented as you requested.

---

## Author Response (AR1)

Jennifer Cunningham
Department of Energy Resources
University of Stavanger
4036, Stavanger
Norway

Dr. CharLotte Krawczyk,

Our manuscript entitled "The impact of seismic interpretation methods on the analysis of faults: A case study from the Snøhvit Field, Barents Sea" has now undergone the public review and we were delighted to receive such constructive and informative reviews by experts in the field of fault analysis.

All of the suggested changes and improvements to the manuscript and figures have been implemented with the exception of two.

- In the methods it was suggested that all references to the time taken for each seismic interpretation should be moved to results. Since we feel this very small aspect of the interpretation would disrupt the flow of the paper, we have left it in the methods section.
- There was a comment suggesting there should not be too much "own-citation" in the introduction. Since we only reference papers from our group that are relevant to the topic, we have opted to leave them in. Also, new key references from other groups make this section more balanced. The papers we are referring to are:
  - Cunningham, J., Cardozo, N., Townsend, C., Iacopini, D. and Wærum, G. O.: Fault deformation, seismic amplitude and unsupervised fault facies analysis: Snøhvit Field, Barents Sea, J. Struct. Geol., 118, 165–180, doi:10.1016/j.jsg.2018.10.010, 2019.
  - Cunningham, J., Cardozo, N., Weibull, W. W. and Iacopini, D.: Investigating the seismic imaging of faults using PS data from the Snøhvit field, Barents Sea and forward seismic modelling (in review at Petroleum Geoscience)

The updated manuscript and figures have now been submitted to Solid Earth. We look forward to hearing from you in due course

Yours sincerely,

**Jennifer Cunningham, Nestor Cardozo, Chris Townsend and Richard Callow**